# MOTION: Multi-Sculpt Evolutionary Coarsening for Federated Continual Graph Learning

**Guancheng Wan**[1†]**, Fengyuan Ran**[1†]**, Ruikang Zhang**[2†]**, Wenke Huang**[1]**,**
**Xuankun Rong**[1]**, Guibin Zhang**[2]**, Yuxin Wu**[3]**, Bo Du**[1*]**, Mang Ye**[1*]

[1]Wuhan University    [2]Tongji University    [3]Renmin University of China
{guanchengwan, yemang}@whu.edu.cn

## Abstract

Graph neural networks (GNNs) have achieved remarkable success in various domains but typically rely on centralized, static graphs, which limits their applicability in distributed, evolving environments. To address this limitation, we define the task of Federated Continual Graph Learning (FCGL), a paradigm for incremental learning on dynamic graphs distributed across decentralized clients. Existing methods, however, neither preserve graph topology during task transitions nor mitigate parameter conflicts in server-side aggregation. To overcome these challenges, we introduce MOTION, a generalizable FCGL framework that integrates two complementary modules: the Graph Topology-preserving Multi-Sculpt Coarsening (G-TMSC) module, which maintains the structural integrity of past graphs through a multi-expert, similarity-guided fusion process, and the Graph-Aware Evolving Parameter Adaptive Engine (G-EPAE) module, which refines global model updates by leveraging a topology-sensitive compatibility matrix. Extensive experiments on real-world datasets show that our approach improves average accuracy (AA) by an average of 30% $\uparrow$ over the FedAvg baseline across five datasets while maintaining a negative $\downarrow$ average forgetting (AF) rate, significantly enhancing generalization and robustness under FCGL settings. The code is available for anonymous access at https://github.com/GuanchengWan/MOTION.

## 1 Introduction

Graph neural networks (GNNs) [27, 63] provide a powerful framework for exploiting relational information in graph-structured data for various learning tasks. Their ability to model complex interdependencies between entities has driven notable progress in domains such as recommendation systems [4], social network analysis [45], and digital marketing optimization [19]. However, existing graph learning methods [72, 63, 68] assume centralized data storage, where a single institution collects and manages the evolving graph, which often fails in real-world scenarios. To address these limitations, researchers have integrated federated learning frameworks with GNNs to develop Federated Graph Learning (FGL) [14, 29, 40, 52]. FGL enables efficient learning on distributed graph data and supports knowledge extraction across decentralized sources.

Graph-structured data in modern systems is inherently dynamic, with continuously adding nodes and edges reflecting changing structures and behaviors [11, 25, 69]. Moreover, storing or accessing extensive historical node profiles and topologies is impractical due to client-side storage limits, edge device constraints, and limited database resources. These constraints prevent edge devices from maintaining a complete history of the evolving graph. Consequently, graph models must incrementally incorporate

---

[†] Co-first authors with equal contributions. Author order was determined alphabetically, with each author reserving the right to be listed first.
[*] Corresponding authors.

39th Conference on Neural Information Processing Systems (NeurIPS 2025).

new classes into a growing graph distributed across decentralized clients. We refer to this process as **Federated Continual Graph Learning (FCGL)**, which addresses storage and edge device limitations. Existing FGL methods [7, 31, 62] assume shared graph data across clients, and current continual graph learning (CGL) approaches [70, 6, 75] focus on static or locally evolving graphs within a single client. The core difficulty unique to FCGL lies in the interplay between local CL limitations and federated aggregation challenges, amplified by client heterogeneity in both data and temporal evolution. Therefore, we define the FCGL task as constructing a global model that generalizes effectively to distributed graphs in an incremental learning framework. A key research question emerges:

*H*ow can we design an FGL framework tailored for continuously evolving streaming data?

Existing CGL methods mainly address node- or class-incremental scenarios [69, 74, 13] and often struggle to preserve the inherent graph topology [12, 67, 51], that is, the structural properties of each local graph such as node centrality and subgraph patterns. In addition, methods such as memory replay [75, 58, 42] or parameter regularization [30, 8, 57, 9] must work under the constraint that clients only have partial and changing graph views with limited resources. As a result, preserving topological integrity, namely the consistent maintenance of these structural properties across tasks during continual updates, remains difficult. This difficulty arises because similarity metrics [48, 57] or criteria [26, 3] computed locally often fail to capture important global relational patterns. This limitation raises a critical question: **I) How can we effectively preserve the topological information of previous tasks under FCL constraints?**

In the FGL setting, client-specific updates reflect heterogeneous objectives and graph structures [34, 76, 60], which can lead to conflicting parameter updates during aggregation. Standard averaging strategies [36] may dilute task-specific knowledge and destabilize the global model, reducing its ability to accommodate new tasks without forgetting previous ones. Although several advanced aggregation methods, such as clustered FL [43], attentive aggregation [23], and weight regularization [1], have been proposed to mitigate client heterogeneity, they still fail to resolve the fundamental issue of update conflicts when task distributions or graph structures diverge substantially [22, 20]. This observation leads to a second question: **II) How can we capture and leverage correlations between the existing parameters of the global model and the client-updated parameters when learning new tasks?**

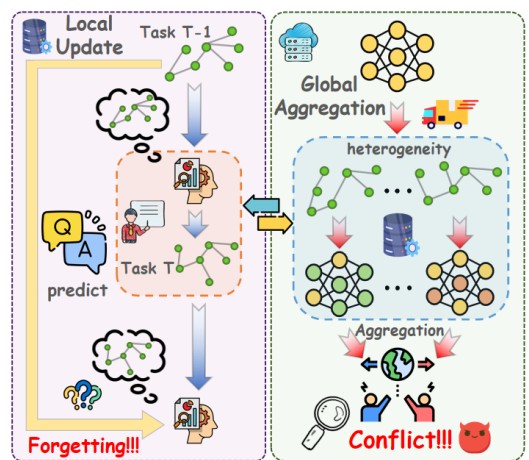

Figure 1: **Problem Illustration**. We describe the challenges FCGL encounters. Local updates cause catastrophic forgetting, while global aggregation creates parameter conflicts.

To address these challenges, we present the first comprehensive study of FCGL. We propose the **M**ulti-Sculpt Ev**O**lu**TION**ary Coarsening (`MOTION`) framework for FCGL. To address Problem **I)**, we develop **Graph Topology-preserving Multi-Sculpt Coarsening (G-TMSC)**. This method mitigates forgetting by preserving the key topological structures of previous task graphs. Inspired by a multi-expert paradigm, we introduce a scoring mechanism that computes node similarity and importance to guide the coarsening process. This mechanism enhances both feature and topology replay, allowing clients to retain critical information from earlier tasks. On the server side, to address Problem **II)**, we propose the **Graph-Aware Evolving Parameter Adaptive Engine (G-EPAE)**. Each client generates topology-sensitive parameter increments that reflect local graph evolution. The server constructs a graph compatibility matrix to measure the alignment between these increments and the global model across parameter dimensions. Aggregation rates are then adjusted dynamically so that increments with high compatibility receive greater weight to capture essential patterns while those with low compatibility are down-weighted to reduce conflicts. All adjusted increments are integrated into the global model without fine-tuning dynamically. This approach achieves balanced evolution that adapts to new structures while preserving historical knowledge securely.

Our principal contributions are summarized as follows:

❶ *Problem Identification.* We formally characterize the main challenges in FCGL: maintaining graph-topological integrity and preserving task-relevant information during CL, while avoiding conflicts in knowledge transfer to ensure robust global generalization.

❷ *Practical Solution.* We design a client-side G-TMSC guided by learned similarity scores to merge new and historical subgraphs, and we develop a server-side G-EPAE that dynamically adjusts aggregation weights to reduce interference and improve coherence.

❸ *Experimental Validation.* We conduct comprehensive experiments on multiple benchmark graph datasets to demonstrate that `MOTION` outperforms existing methods under FCGL settings.

## 2 Preliminaries

### 2.1 Notations

**Continual Learning.** CL involves updating a model sequentially on a series of tasks, where each task introduces new information while requiring the retention of knowledge from previous tasks. The primary objective is to mitigate catastrophic forgetting, in which the model's performance on earlier tasks deteriorates as it learns new ones. Formally, let the model encounter a sequence of task-specific datasets $D_t = \{X_t, Y_t\}$, where $X_t$ denotes the input data and $Y_t$ the corresponding labels for task $t$. The model parameters $\theta$ are updated by balancing the acquisition of new information with the preservation of prior knowledge. This trade-off can be expressed as:

$$\mathcal{P}_{\text{total}}(\theta) = \sum_{t=1}^{T} \mathcal{P}_t(\theta) + \lambda \, \mathcal{R}(\theta), \tag{1}$$

where $\mathcal{P}_t(\theta)$ represents the update term for task $t$, $\mathcal{R}(\theta)$ serves as a regularization term that preserves knowledge from previously learned tasks, and the hyperparameter $\lambda$ modulates the trade-off between learning new tasks and reducing catastrophic forgetting.

In practice, only the data from the current task $D_T$ is accessible during training. Therefore, the objective at step $T$ is formulated as:

$$\mathcal{L}_T(\theta) = \mathcal{P}_T(\theta) + \lambda_T \mathcal{R}(\theta), \tag{2}$$

which highlights the central challenge in CL, namely achieving high performance on the current task while preserving knowledge acquired from earlier tasks through regularization.

**Problem Formulation.** In the FCGL framework, a central server coordinates distributed learning across $K$ clients, denoted $\mathcal{C} = \{C_1, \ldots, C_K\}$. Each client $C_k$ maintains an evolving graph $G_k^t = (V_k^t, E_k^t)$ for task $t$, with adjacency matrix $A_k^t$. Each node $v_i \in V_k^t$ has an associated feature vector $x_i^{k,t}$ and label $y_i^{k,t}$. The goal is to learn new tasks incrementally while preserving prior knowledge to ensure the stability of the global model. To this end, each client $k$ trains a local model $\mathcal{F}_{\theta_k^t}$ with parameters $\theta_k^t$ and periodically transmits its updates to the central server. The global model parameters $\phi$ are optimized by minimizing the weighted sum of local losses:

$$\min_{\phi} \sum_{k=1}^{K} \frac{N_k^t}{\mathbb{N}_t} L_k^t(\phi), \tag{3}$$

where $N_k^t = |V_k^t|$ is the number of nodes in the graph of client $k$ at task $t$, $\mathbb{N}_t = \sum_{k=1}^{K} N_k^t$ is the total number of nodes across all clients at task $t$, and $L_k^t(\phi)$ denotes the expected loss over $G_k^t$ under the global model parameters $\phi$. FCGL unifies FCL and CGL by enabling incremental task learning and knowledge preservation while protecting data privacy through distributed model aggregation.

### 2.2 Motivation

This paper systematically examines the challenges of maximizing the generalization performance of the global model in FCGL. Traditional FCL and CGL methods, such as Experience Replay [75] and Gradient Episodic Memory [33], were designed for image-based tasks and do not deliver comparable performance on graph-structured data. These methods suffer from three main shortcomings. First, they cannot effectively capture graph topological information [56, 32, 54]. Second, they require excessive storage [58, 75] for node replay buffers. Third, they introduce severe parameter conflicts

during aggregation, as heterogeneous local updates push the global model in divergent directions [55, 53, 49]. By relying on simple empirical node replay, these methods ignore the structural relationships that encode critical topological patterns. This increases misalignment between client and server, destabilizes training, and accelerates catastrophic forgetting. As shown in Figure 1, nodes exhibit substantial heterogeneity in connectivity patterns and neighborhood distributions. Therefore, local training in FCGL must minimize storage requirements while preserving graph topology and node information to support incremental learning without loss of prior knowledge. We formalize this objective as:

$$\{\theta_k^{t*}, M_k^{t*}\} = \arg \min_{\theta_k^t, M_k^t} \sum_{k=1}^{K} \Big[ \mathbb{E}_{(x,y)\sim D_t^k} \ell\big(F_{\theta_k^t}(x), y\big) + \lambda \, \mathcal{L}_{\mathrm{ret}}\big(\theta_k^t; G_k^{1:t-1}\big) \Big] \quad \text{s.t.} \quad |M_k^t| \le B,$$
(4)

where $\ell$ is the per-sample classification loss, $\mathcal{L}_{\mathrm{ret}}$ enforces topology-preserving replay, $\lambda$ balances new-task learning and forgetting, and $B$ bounds each client's replay memory capacity.

After local training effectively mitigates knowledge forgetting, we subsequently integrate client-trained parameters into the global model $F_\phi$ during the server-side aggregation phase, ensuring a conflict-free aggregation process that faithfully preserves the knowledge previously acquired:

$$\phi^* = \arg \min_\phi \sum_{k=1}^{K} \mathbb{E}_{\hat{g}_k \sim R_\varphi^k} \Big[ T\big(F_\phi(\hat{g}_k), \, F_{\theta_k^t}(\hat{g}_k)\big) \Big],$$
(5)

where $R_\varphi^k$ produces replay samples for client $k$, and $T(\cdot, \cdot)$ measures the discrepancy between global and local model outputs. This approach ensures retention of previously accumulated knowledge.

Based on these objectives, we therefore comprehensively outline the systematically derived key design principles for a generalizable FCGL pipeline:

> **Generalizable FCGL Design Principles**: *Storage Efficiency: preserve critical topological structures and key node information while reducing memory requirements; **Knowledge Retention**: track changes in graph topology and node information across tasks without losing previously learned knowledge; **Knowledge Integration**: aggregate client updates into the global model with minimal conflicts to ensure effective knowledge transfer and mitigate forgetting.*

In the subsequent sections, we describe how our approach effectively implements these design principles. Specifically, we minimize client-side storage, prevent catastrophic knowledge loss, and seamlessly integrate updates into the global model with reduced parameter conflicts.

## 3 Methodology

### 3.1 Framework Overview

In this section, we briefly present an overview of our framework. On the client side, we apply G-TMSC to accurately preserve the critical topological structures from previous task graphs. G-TMSC uses a multi-expert scoring mechanism to estimate node similarity precisely and guide structural fusion. On the server side, we employ G-EPAE to mitigate conflicts in the task space during aggregation. G-EPAE adjusts parameter aggregation rates dynamically based on a graph compatibility matrix that measures the alignment between client updates and the global model. Moreover, a diagrammatic representation of the framework is provided in Figure 2.

### 3.2 Graph Topology-preserving Multi-Sculpt Coarsening (G-TMSC)

**Dynamic Graph Merging.** To integrate graph data across successive tasks, we introduce Dynamic Graph Merging, which constructs a progressively evolving knowledge graph. For client $k$ at task $t$, let $G_k^t = (V_k^t, E_k^t)$ be the local graph. Each node $v_i \in V_k^t$ is assigned a unique global identifier by fusing its semantic feature $x_i^{k,t}$ with the task identifier $task_t$:

$$\hat{z}_i = \mathrm{Fuse}(x_i^{k,t}, \, task_t),$$
(6)

where $\mathrm{Fuse}(\cdot)$ denotes a task-aware embedding fusion module. All global identifiers are stored in a centralized node mapping table $\mathcal{I}_G$ to ensure consistent entity referencing across tasks. To prevent

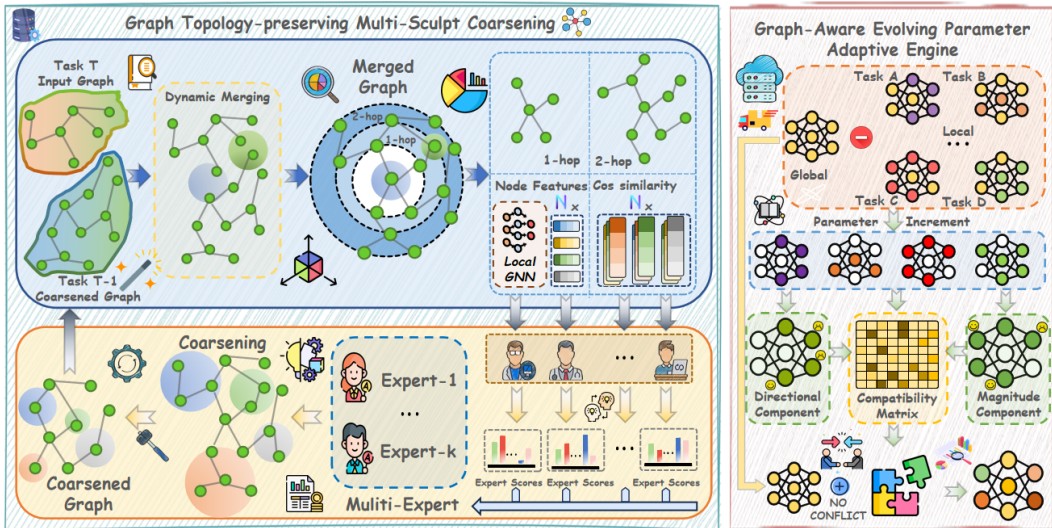

Figure 2: Architecture illustration of `MOTION`. (a) *The left part* shows the Graph Topology-preserving Multi-Sculpt Coarseing (G-TMSC) on the client side. (b) *The right part* presents Graph-Aware Evolving Parameter Adaptive Engine (G-EPAE) on the server side.

identity drift across tasks, we maintain two mappings: a many-to-one mapping $\mathcal{M}_{\text{coarse}\rightarrow\text{original}}$ from merged to original nodes, and a one-to-one mapping $\mathcal{M}_{\text{original}\rightarrow\text{coarse}}$ for traceability. This dual mapping supports dynamic node aggregation while preserving entity fidelity.

At initialization ($t = 1$), we construct a base graph using the node and edge mappings of $G_k^1$. For each subsequent task $t = 2, \ldots, T$, we apply an incremental merging strategy: unseen nodes $v_j \notin \mathcal{I}_G$ are identified by global hashing and assigned embeddings $\hat{z}_j$. This selective update avoids exponential growth in computation from naive accumulation of all previous graphs and improves scalability. Thus, each client $k$ refines its evolving graph $G_k^t$, capturing long-range dependencies via coarsening and mapping, and progressively unifies knowledge across tasks.

For edges $E_k^t$, we transform the local adjacency matrix $A_k^t$ into a unified matrix $\hat{A}_k^t$ defined by

$$\hat{A}_k^t(i,j) = \begin{cases} 1, & \text{if } (v_i, v_j) \in E_k^t \text{ and } v_i, v_j \in \mathcal{I}_G, \\ 0, & \text{otherwise,} \end{cases} \tag{7}$$

which preserves structural integrity while reducing redundant computation overhead. Therefore, task-specific representations are continuously integrated with explicit preservation of graph topology and node semantics, ensuring historical knowledge retention and new information assimilation.

**Multi-Sculpt Coarsening.** Sculpting in graph coarsening refers to the precise selective removal of non-essential or peripheral elements to preserve the fundamental core structural integrity of a graph. Inspired by the Mixture-of-Experts (MoE) paradigm, our approach employs a collaborative multi-expert framework in which each expert specifically applies a distinct evaluation criterion, such as topological centrality, local substructure decomposition, and semantic similarity. By integrating these expert assessments, the Multi-Sculpt Coarsening framework systematically prunes extraneous redundant nodes and suppresses noise, thereby significantly reducing storage and computational demands while preserving semantic fidelity and essential graph properties.

First, we compute a diverse set of topological features for each node. Degree centrality quantifies immediate connectivity: $\text{Degree}(v_i) = \sum_{v_j \in V_k^t} A_k^t(i,j)$, while betweenness centrality identifies critical bridges: $\text{BC}(v_i) = \sum_{s \neq i \neq t} \frac{\sigma_{st}^{(k,t)}(v_i)}{\sigma_{st}^{(k,t)}}$. The clustering coefficient further measures community compactness via $C(v_i) = \frac{2 T_i^{(k,t)}}{k_i (k_i-1)}$, where $T_i^{(k,t)}$ is the number of triangles through $v_i$ and $k_i = \text{Degree}(v_i)$. Additionally, global metrics such as eigenvector centrality and closeness centrality further enrich the feature set. To model diffusion, we apply PageRank:

$$PR(v_i) = \frac{1-d}{N_k^t} + d \sum_{v_j \in \mathcal{N}(v_i)} \frac{PR(v_j)}{\text{Degree}(v_j)}, \tag{8}$$

with damping factor $d$ and neighborhood $\mathcal{N}(v_i)$. We also derive degree-based heterogeneity and a community index to capture neighborhood diversity and cohesion.

For fine-grained analysis, we decompose the graph into 1-hop and 2-hop subgraphs. We compute the average degree $\bar{k}_{\text{sub}} = \frac{1}{|V_{\text{sub}}|} \sum_{v \in V_{\text{sub}}} k_v$ and measure subgraph diameter $D_{\text{sub}}$. Subgraph clustering coefficients and densities further characterize community cohesiveness.To comprehensively represent each node, we introduce a position-aware degree embedding scheme employing multi-frequency sinusoidal encodings to capture structural variations:

$$\text{PE}_k(d) = \begin{cases} \sin\big(d/10000^{2k/F}\big), & k \text{ even}, \\ \cos\big(d/10000^{2k/F}\big), & k \text{ odd}, \end{cases} \tag{9}$$

where $d$ is the node degree and $F$ is the embedding dimension.

In our proposed framework, the similarity assessment module employs statistical metrics. Maximum mean discrepancy (MMD) detects distributional shifts:

$$\text{MMD}^2(X, Y) = \left\| \frac{1}{m} \sum_{i=1}^{m} \phi(x_i) - \frac{1}{n} \sum_{j=1}^{n} \phi(y_j) \right\|^2, \tag{10}$$

while Mahalanobis distance identifies statistical outliers via $D_M(x) = \sqrt{(x - \mu)^\top \Sigma^{-1} (x - \mu)}$. We subsequently compute Pearson correlation and cosine similarity metric via $\text{CosSim}(x, y) = \frac{x \cdot y}{\|x\|\|y\|}$, to capture both functional and semantic alignments.

Based on these features, we implement a multi-expert decision mechanism where each expert specializes in a distinct feature domain. Employing a sparse activation strategy selects the top-$K$ experts dynamically, and introduces controlled random noise perturbation along with the coefficient of variation squared $\text{CV}^2 = \frac{\sigma^2}{\mu^2}$ to ensure balanced and equitable expert utilization. Subsequently, we aggregate experts' normalized importance scores to prioritize nodes for preservation, then map removed nodes to their semantically closest preserved counterparts to minimize information loss.

To merge features in the coarsened graph, we apply simple yet robust arithmetic averaging at both the raw feature and respective latent representation levels:

$$x_{\text{merged}} = \frac{1}{n} \sum_{i=1}^{n} x_i, \quad h_{\text{merged}} = \frac{1}{n} \sum_{i=1}^{n} h_i, \tag{11}$$

ensuring unbiased aggregation and retention of semantic and structural integrity.

Finally, we introduce a reservoir sampling–based dynamic memory mechanism that blends selected nodes from historical tasks with new samples. This mechanism mitigates gradient shocks, stabilizes convergence, and enhances robustness by regulating the mixing ratio between past and new nodes.

### 3.3 Graph-Aware Evolving Parameter Adaptive Engine (G-EPAE)

**Graph Compatibility Matrix.** Upon completion of local training, each client $k \in \mathcal{C}$ transmits its updated parameters $\theta^k$ to the central server. The server then computes a topology-sensitive parameter increment: $\Delta\theta^k = \theta^k - \phi$, where $\phi$ denotes the current global parameters of $\mathcal{F}_\phi$. This increment implicitly captures structural changes in the local graph $G_k^t = (V_k^t, E_k^t)$, since GNN parameter updates reflect variations in the adjacency matrix $A_k^t$ through neighborhood aggregation.

To integrate these heterogeneous updates, we introduce a graph compatibility matrix $\mathbf{M}_k \in \mathbb{R}^{d \times d}$ that quantifies the alignment between local and global update dynamics. Unlike traditional aggregation methods (e.g., FedAvg), which weight updates solely by data volume, our matrix incorporates both directional consistency and relative magnitude of each client's update:

$$\mathbf{M}_k = \Delta\theta_{\text{dir}}^k \circ \Delta\theta_{\text{mag}}^k, \tag{12}$$

where $\circ$ denotes the Hadamard product. The directional component is obtained by L2 normalization:

$$\Delta\theta_{\text{dir}}^k = \frac{\Delta\theta^k}{\|\Delta\theta^k\|_2}, \tag{13}$$

which isolates the update direction from its scale. The magnitude component is defined as:

$$\Delta\theta_{\text{mag}}^k = \frac{\|\Delta\theta^k\|_2}{\sum_{j\in\mathcal{C}}\|\Delta\theta^j\|_2}, \tag{14}$$

providing a normalized importance score that emphasizes clients with larger topological contributions.

By incorporating $\mathbf{M}_k$ into the aggregation process, we enable a topology-aware and contribution-sensitive update mechanism. The directional component $\Delta\theta_{\text{dir}}^k$ ensures that the global model aligns more closely with structurally consistent client updates, which reduces the cancellation effect caused by conflicting update directions. At the same time, the magnitude component $\Delta\theta_{\text{mag}}^k$ emphasizes clients whose updates reflect more substantial topological shifts in their local graphs. The Hadamard combination then produces a compatibility matrix that softly reweights each update. This conflict-aware integration retains essential task-specific knowledge and improves global model stability.

**Elastic Regulation Fusion.** Building on compatibility-guided aggregation principle, we introduce an elastic regulation fusion mechanism to further enhance both adaptability and long-term stability in global model synchronization for FCGL. This mechanism applies a dynamic gating function $g(\mathbf{M}_k)$ to appropriately scale aggregation weights nonlinearly:

$$\alpha_k = \begin{cases} \exp(\mathbf{M}_k), & \mathbf{M}_k \geq \tau, \\ \gamma\,\mathbf{M}_k, & \text{otherwise}, \end{cases} \tag{15}$$

where $\tau$ is the compatibility threshold, $\gamma \in (0,1)$ is a linear decay factor, and $\alpha_k$ regulates the contribution of each local update $\Delta\theta^k$ to the global model. To suppress instability from incompatible updates, we introduce an adaptive regularization term:

$$\mathcal{R}(\theta) = \lambda\big\|\Delta\theta_d\big\|_2^2, \quad \Delta\theta_d = \{\,\Delta\theta_i^k \mid \mathbf{M}_k(i) < \tau\,\}, \tag{16}$$

which selectively truncates gradients of poor compatibility parameters. The global model is then updated by integrating the filtered and scaled local updates:

$$\Delta\phi = \sum_{k=1}^{K} \alpha_k\,\Delta\theta^k, \quad \phi \leftarrow \phi + \eta\,\Delta\phi. \tag{17}$$

This update enables $\mathcal{F}_\phi$ to assimilate emerging structural patterns from clients while preserving representations learned in previous tasks. From a mathematical perspective, the fusion strategy performs adaptive filtering in parameter space: tensor level aggregation is jointly governed by directional alignment via normalized updates $\Delta\theta_{\text{dir}}^k$ and compatibility driven scaling $\alpha_k$, allowing fine-grained, client specific fusion across both parameter dimensions and client axes.

The proposed method offers two primary benefits in FCGL. First, it enhances the responsiveness of the model to dynamic topological changes through $\mathbf{M}_k$-guided integration. Second, it ensures minimal conflict during parameter evolution across tasks. These benefits collectively enable robust and scalable continual learning in dynamic graph environments.

## 4 Experiment

In this section, we comprehensively evaluate MOTION through four axes: **Q1** (Superiority), **Q2** (Resilience), **Q3** (Effectiveness), **Q4** (Sensitivity).

### 4.1 Experimental Setup

**Datasets.** To effectively evaluate the performance of our approach, we employed five benchmark graph datasets of various scales and distributions, including Cora [35], CiteSeer [16], PubMed [5], Amazon-Photo, and Coauthor-CS [44]. Detailed descriptions and dataset splits are provided in Appendix C.1. Moreover, implementation details and parameter settings are given in Appendix C.6.

**Counterparts.** We compare MOTION against the following representative baselines covering classical FL, FGL, FCL, and CGL methods: (1) **FedAvg** [ASTATS17] [36], (2) **FedDc** [CVPR22] [15], (3) **FedDyn** [ICLR21] [2], (4) **FedSSL** [IJCAI24] [21], (5) **FedSSP**[NeurIPS24] [50], (6) **FedTpp** [ICML24] [41], (7)**SEA-ER** [CoLLAs24][47], and (8) **FedPowde** [NeurIPS24] [38]. Detailed descriptions are provided in Appendix C.2.

Table 1: **Comparison with the state-of-the-art methods** on five real-world datasets. We report node-classification Average Accuracy (AA) (%) and Average Forgetting (AF) (%) for our downstream tasks. Green indicate improvements over the FedAvg baseline, while red denote performance declines. Meanwhile, ↑ denotes an increase in the value, whereas ↓ denotes a decrease. The best and second results are highlighted with **bold** and underline, respectively.

| Category | Methods | Cora | | CiteSeer | | PubMed | | Amz-Photo | | Coauthor-CS | |
|---|---|---|---|---|---|---|---|---|---|---|---|
| | | $AA \Uparrow$ | $AF \Downarrow$ | $AA \Uparrow$ | $AF \Downarrow$ | $AA \Uparrow$ | $AF \Downarrow$ | $AA \Uparrow$ | $AF \Downarrow$ | $AA \Uparrow$ | $AF \Downarrow$ |
| FL | FedAvg [ASTATS17] | 23.57 | 56.30 | 16.67 | 69.01 | 33.33 | 50.42 | 18.43 | 54.74 | 12.15 | 65.78 |
| | FedDc [CVPR22] | 25.64↑2.07 | 58.32↑2.02 | 21.12↑4.45 | 43.60↓25.41 | 33.25↓0.08 | 50.00↓0.42 | 21.08↑2.65 | 42.22↓12.52 | 10.42↓1.73 | 49.79↓15.99 |
| | FedDyn [ICLR21] | 18.93↓4.64 | 71.07↑14.77 | 13.59↓3.08 | 67.52↓1.49 | 26.19↓7.14 | 55.44↑5.02 | 18.48↑0.05 | 36.32↓18.42 | 9.19↓2.96 | 45.19↓20.59 |
| FGL | FedSSL [IJCAI24] | 20.14↓3.43 | 29.27↓27.03 | 15.80↓0.87 | 39.14↓29.87 | 33.34↑0.01 | 25.08↓25.34 | 32.31↑13.88 | 9.76↓44.98 | 7.28↓4.87 | 14.77↓51.01 |
| | FedSSP [NeurIPS24] | 25.76↑2.19 | 26.60↓29.70 | 21.83↑5.16 | 16.11↓52.90 | 38.54↑5.21 | 31.36↓19.06 | 14.09↓4.34 | 13.39↓41.35 | 10.22↓1.93 | 14.10↓51.68 |
| CGL | FedTpp [ICML24] | 53.63↑30.06 | 19.88↓36.42 | 27.67↑11.00 | 55.80↓13.21 | 49.02↑15.69 | 3.09↓47.33 | 29.75↑11.32 | 38.22↓16.52 | 19.15↑7.00 | 58.28↓7.50 |
| | SEA-ER [CoLLAs24] | 20.05↓3.52 | 69.17↑12.87 | 31.38↑14.71 | 47.26↓21.75 | 55.79↑22.46 | 1.77↓48.65 | 15.21↓3.22 | 48.93↓5.81 | 6.88↓5.27 | 82.62↑16.84 |
| FCL | FedPowde [NeurIPS24] | 19.05↓4.52 | 71.32↑15.02 | 20.93↑4.26 | 47.99↓21.02 | 33.42↑0.09 | 49.87↓0.55 | 22.73↑4.30 | 50.85↓3.89 | 10.39↓1.76 | 64.09↓1.69 |
| FCGL | MOTION | **62.66**↑39.09 | **−8.34**↓64.64 | **49.08**↑32.41 | **−16.78**↓85.79 | **59.48**↑26.15 | **−24.17**↓74.59 | **79.09**↑60.66 | **−5.32**↓60.06 | **22.26**↑10.11 | **−11.62**↓77.40 |

## 4.2 Superiority

To address **Q1**, we analyze the superior performance of `MOTION` on node classification across multiple real-world graph datasets. The overall Average Accuracy (AA↑) and Average Forgetting (AF↓) results are summarized in Table-1. Here, $AA = \frac{1}{T}\sum_{j=1}^{T} a_{T,j}$, $AF = \frac{1}{T}\sum_{j=1}^{T}\left(\max_{l \in \{1,...,T\}} a_{l,j} - a_{T,j}\right)$, where $T$ is the total number of tasks and $a_{i,j}$ is the performance of the global model on task $j$ after training on task $i$. From these experiments, we derive three key observations.

Obs.❶ `MOTION` consistently achieves the highest accuracy on all evaluated datasets, demonstrating notably higher AA and negative AF values. Negative AF indicates not only preservation but also reinforcement of prior knowledge. On Amazon-Photo, `MOTION` outperforms the second-best baseline, FedTPP, by 48.34% in AA and yields a negative AF, highlighting its strong capabilities for knowledge consolidation and conflict mitigation. The G-TMSC module preserves critical topological patterns, and the G-EPAE module reduces parameter conflicts between clients and server. Together, these components enable `MOTION` to capture fine-grained structural knowledge across heterogeneous tasks.

Obs.❷ Traditional FL and FGL methods tend to overfit to the current task, which leads to catastrophic forgetting and prevents clients from retaining prior knowledge. Moreover, the server encounters significant difficulty in resolving parameter conflicts, resulting in substantial performance degradation.

Obs.❸ Both FCL and CGL approaches struggle under the FCGL paradigm. CGL methods cannot resolve server-side parameter conflicts, and FCL techniques fail to capture essential topological structures. Among the baselines, FedTPP and FedPowde deliver the next-best AA; however, FedTPP still exhibits high AF due to aggregation-induced conflicts, and FedPowde's weaker structural modeling leads to lower AA and increased forgetting. Consequently, neither method supports efficient, continuous knowledge sharing across heterogeneous clients.

## 4.3 Resilience

To address **Q2**, we evaluate `MOTION` on the Cora and CiteSeer datasets. We vary the Dirichlet concentration parameter $\alpha$ from 1 to 9 to control data heterogeneity and assess performance with 2 to 6 clients. Figure 3 shows that `MOTION` achieves robust gains across all client scales and $\alpha$ values. On CiteSeer, `MOTION` outperforms FedAvg by an average of 29.11% and exceeds the strongest competing baseline by at least 18.11%. These findings confirm that `MOTION` adapts effectively to diverse FCGL scenarios and maintains stable performance under severe client heterogeneity.

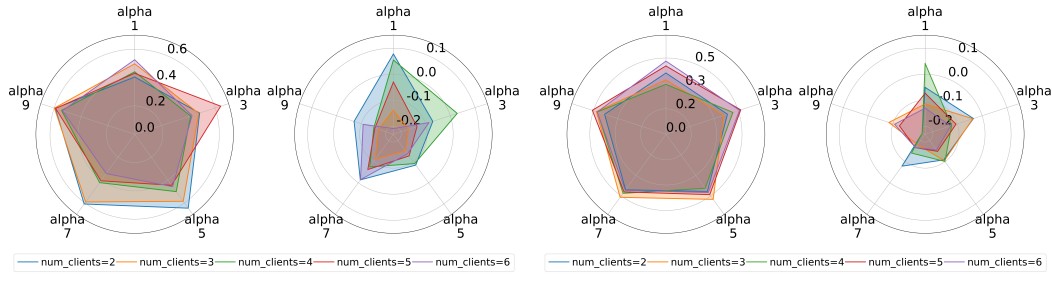

(a) Analysis on Cora            (b) Analysis on CiteSeer

Figure 3: **Resilience study** of hyperparameters Reduction Rate and Experts Selected with datasets including Cora and CiteSeer. Please refer to Sec. 4.3 for further analysis.

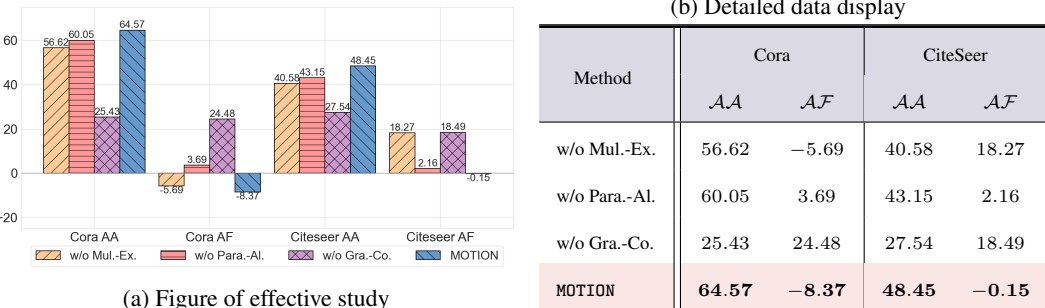

(b) Detailed data display

| Method | Cora | | CiteSeer | |
|---|---|---|---|---|
| | $\mathcal{AA}$ | $\mathcal{AF}$ | $\mathcal{AA}$ | $\mathcal{AF}$ |
| w/o Mul.-Ex. | 56.62 | $-5.69$ | 40.58 | 18.27 |
| w/o Para.-Al. | 60.05 | 3.69 | 43.15 | 2.16 |
| w/o Gra.-Co. | 25.43 | 24.48 | 27.54 | 18.49 |
| MOTION | **64.57** | $-8.37$ | **48.45** | $-0.15$ |

(a) Figure of effective study

Figure 4: **Effectiveness study** of G-TMSC and G-EPAE on Cora and CiteSeer. For an in-depth study, see Sec. 4.4.

## 4.4 Effectiveness

To address **Q3**, we conduct an ablation study to evaluate the contributions of key components on both client and server sides. In Figure 4, we first isolate the effects of the Graph-Coarsening mechanism (w/o Gra.-Co.) and the Multi-Expert strategy (w/o Mul.-Ex.). Removing the Graph-Coarsening mechanism alone causes a significant performance decrease, and further exclusion of the Multi-Expert strategy amplifies this decline. These results demonstrate that both components are crucial for capturing diverse structural patterns and mitigating knowledge forgetting. We then assess the Parameter-Alignment mechanism (w/o Para.-Al.) within the G-EPAE module. Its absence intensifies client–server parameter conflicts and increases update drift, confirming its essential role in harmonizing model updates and enabling continuous knowledge accumulation during FCGL.

## 4.5 Sensitivity

To investigate hyperparameter sensitivity for **Q4**, we conduct a systematic study on `MOTION`. As shown in Figure 5, we examine two key parameters: the graph reduction rate $r$ and the number of selected experts $s$. We vary $r$ from 0.1 to 0.9 in increments of 0.2 and adjust $s$ from 2 to 10 in steps of 2. The results show that model performance varies modestly across this parameter range, indicating strong robustness. On Cora, AA and AF exhibit variances of 0.49 and 1.84, respectively. On CiteSeer, the variances are 0.70 for AA and 0.44 for AF. To balance compression and information retention, we set $r=0.5$ and $s = 3$, which promotes effective multi-expert collaboration while minimizing storage.

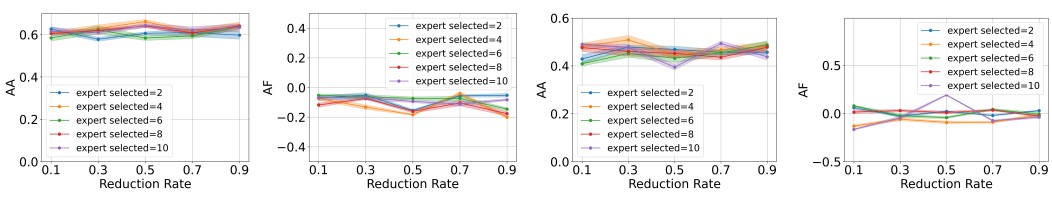

(a) Analysis on Cora            (b) Analysis on CiteSeer

Figure 5: **Sensitivity Study** of hyperparameters Reduction Rate and Experts Selected. Please refer to Sec. 4.5 for further analysis.

# 5 Conclusion

In this paper, we propose `MOTION`, a unified FCGL framework that simultaneously preserves historical graph structures and mitigates aggregation conflicts in decentralized, evolving environments. On the client side, our G-TMSC module integrates multiple structural metrics through a multi-sculpt scheme inspired by MOE, in which each expert applies a distinct evaluation criterion and the results are fused via similarity-guided merging. On the server side, our G-EPAE module adjusts aggregation weights based on a topology-sensitive compatibility matrix to align heterogeneous client updates while maintaining task-specific knowledge. Extensive experiments on five real world graph benchmarks demonstrate that `MOTION` consistently outperforms state-of-the-art baselines in both generalization performance and robustness to catastrophic forgetting. Comprehensive empirical evaluations across diverse datasets validate the robustness and efficacy of `MOTION`.

# Acknowledgement

This work is supported by National Natural Science Foundation of China under Grant (62361166629, 62225113, 623B2080), the Major Project of Science and Technology Innovation of Hubei Province (2024BCA003, 2025BEA002), and the Innovative Research Group Project of Hubei Province under Grants 2024AFA017. The supercomputing system at the Supercomputing Center of Wuhan University supported the numerical calculations in this paper.

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

# A   Notations

We present a comprehensive review of the commonly used notations and their definitions in Tab. 2.

Table 2: Notation and Definitions

| Notation | Definition |
|---|---|
| $\mathcal{G}$ | Graph data. |
| $\mathcal{V}$ | The node set of $\mathcal{G}$. |
| $\mathcal{E}$ | The edge set of $\mathcal{G}$. |
| $D_t$ | Task-specific dataset for task $t$. |
| $X_t$ | Input data for task $t$. |
| $Y_t$ | Lables corresponding to the input data $X_t$ for task $t$. |
| $T$ | The number of tasks. |
| $K$ | The number of clients. |
| $\theta^k$ | The parameters of the local model $\mathcal{F}_{\theta^k}$ of client $k$. |
| $\mathcal{F}_\phi$ | The global model. |
| $\phi$ | The parameters of the global model. |
| $\mathcal{P}_t(\theta)$ | The parameter update for task $t$. |
| $\mathcal{R}(\theta)$ | The regularization term for parameters updated in CL. |
| $\lambda$ | Trade-off hyperparameter. |
| $\mathcal{C}$ | The set of all clients. |
| $k$ | The index of a specific client in $\mathcal{C}$. |
| $G_k^t$ | The evolving graph maintained by client $k$ for task $t$. |
| $A_k^t$ | The adjacency matrix corresponding to the graph $G_k^t$. |
| $v_i$ | A node in the graph $G_k^t$, where $v_i \in V_k^t$. |
| $x_i^{k,t}$ | The feature vector associated with node $v_i$ in client $k$'s graph for task $t$. |
| $y_i^{k,t}$ | The label associated with node $v_i$ in client $k$'s graph for task $t$. |
| $\mathcal{F}_{\theta_k^t}$ | The local model trained by client $k$ for task $t$. |
| $\theta_k^t$ | The parameters of the local model $\mathcal{F}_{\theta_k^t}$ of client $k$ for task $t$. |
| $\phi$ | The parameters of the global model shared across all clients. |
| $L_k^t(\cdot)$ | The local objective function for client $k$'s graph $G_k^t$ at task $t$. |
| $N_k^t$ | The number of nodes in client $k$'s graph $G_k^t$ at task $t$. |
| $\mathbb{N}_t$ | The total number of nodes across all clients' graphs for task $t$. |
| $\hat{z}_i$ | Global unique ID for node $v_i$, fusing $x_i^{k,t}$ and $task_t$. |
| $\mathcal{I}_G$ | Centralized table indexing global IDs for consistent entity referencing. |
| $\mathcal{M}_{\text{coarse}\rightarrow\text{original}}$ | Many-to-one mapping from merged to original nodes. |
| $\mathcal{M}_{\text{original}\rightarrow\text{coarse}}$ | One-to-one mapping preserving traceability to merged nodes. |
| $\sigma_{st}^{(k,t)}$ | From $s$ to $t$ shortest paths in $G_k^t$. |
| $T_i^{(k,t)}$ | Number of triangles through $v_i$. |
| $PR(v_i)$ | PageRank score of $v_i$. |
| $\mathcal{N}(v_i)$ | Neighbors of $v_i$. |
| $\mathbf{M}_k$: | Graph compatibility matrix for client $k$. |
| $\Delta\theta_{\text{dir}}^k$ | Directional component of client $k$'s update (normalized). |
| $\Delta\theta_{\text{mag}}^k$ | Magnitude-aware component of client $k$'s update. |
| $\alpha_k$ | Aggregation coefficient for client $k$ based on compatibility score |
| $\gamma$ | Linear decay factor for low-compatibility updates |
| $g(\mathbf{M}_k)$ | Dynamic gating function applied to compatibility matrix |
| $\mathbf{M}_k(i)$ | Compatibility value for parameter $i$ from client $k$ |
| $\mathcal{L}_{\text{ret}}(\theta_k^t; G_k^{1:t-1})$ | Replay regularization to preserve prior task knowledge. |
| $B$ | Memory budget constraint for client local replay buffer. |
| $R_\varphi^k$ | Replay buffer for client $k$. |
| $\hat{g}_k$ | A replayed sample from client $k$'s replay buffer. |
| $T(\cdot,\cdot)$ | Discrepancy loss function between two models. |

# B  Related Work

**Federated Continual Learning (FCL).** FCL [65, 46, 73, 71] integrates federated learning with continual learning to enable clients to train on private task streams sequentially while retaining historical knowledge without accessing past data or exchanging raw inputs—thereby mitigating both temporal and spatial catastrophic forgetting. Existing FCL methods generally employ regularization-based techniques, such as FedCurv [66], which adapts Elastic Weight Consolidation to penalize deviations in parameters critical for previous tasks, or rehearsal-based strategies [77, 10] like FedPMR [61], which maintains a compact exemplar buffer and aligns probability distributions via replay.

**Continual Graph Learning (CGL).** CGL [24, 64, 18, 17] addresses the challenge of incrementally updating models on evolving graph-structured data, where nodes, edges or labels may change over time, without reaccessing all past data. Existing CGL methods generally adopt one of two paradigms. Isolation-based approaches, such as Progressive Graph Networks [59], allocate task-specific submodules to prevent interference between new and old tasks. In contrast, replay-based frameworks like GraphReplay [39] maintain a buffer of historical graph snapshots for rehearsal.

**Federated Continual Graph Learning (FCGL) .** FCGL enables decentralized, incremental training on evolving graph-structured data across multiple clients. Existing FGL methods typically assume access to a centralized graph, whereas CGL approaches focus on static or single-client graphs, often sacrificing topological fidelity and exacerbating forgetting through naive parameter averaging. Moreover, the unrefined integration of these two mechanisms erodes task-specific knowledge and introduces parameter conflicts, thereby accelerating structural forgetting and destabilizing the global model. To address these limitations, we propose `MOTION`, the first framework to integrate hierarchical structure-preserving aggregation into FCGL.

# C  Experimental Details.

## C.1  Dataset Details

To evaluate the effectiveness of `MOTION`, we conduct extensive experiments on eight real-world graph datasets spanning multiple domains: the citation networks Cora, CiteSeer, and PubMed; the product co-purchasing network Amazon-Photo; and the academic collaboration network CoAuthor-CS, among others. Each dataset is partitioned into fixed subsets of 20% for training, 40% for validation, and 40% for testing. Table 3 summarizes the key statistics of these datasets. Detailed descriptions follow:

- **Cora/CiteSeer/PubMed.** These standard citation network benchmarks represent scientific publications as nodes and directed citation links as edges. Each paper is described by high-dimensional bag-of-words features derived from its text and labeled according to research topics. Their sparse connectivity and rich feature spaces make them foundational testbeds for node classification and scalability evaluations of GNNs.
- **Amazon-Photo.** Derived from Amazon's co-purchasing data, this network connects photography-related products that are frequently bought together. Nodes encode visual feature descriptors extracted from product images, and the classification task predicts product categories. This dataset challenges models to handle non-textual, image-based features in a commercial recommender context.
- **CoAuthor-CS.** This academic collaboration network links co-authored computer science papers via undirected edges. Node features combine title and abstract embeddings with publication metadata, requiring models to capture interdisciplinary research themes for topic classification. The network's moderate density and heterogeneous feature types test the ability to integrate semantic and structural information.

## C.2  Counterpart Details.

This section presents a detailed overview of the baseline methods employed in our experiments.

- **FedAvg** [ASTATS'17] [36] .The seminal framework for federated learning, which orchestrates synchronous parameter updates through iterative client–server interactions. Each client performs local training epochs on its private data and transmits gradient updates rather than raw inputs to

Table 3: **Statistics** of datasets used in experiments.

| Dataset | #Nodes | #Edges | #Classes | #Features |
|---------|--------|--------|----------|-----------|
| Cora | 2,708 | 5,278 | 7 | 1,433 |
| Citeseer | 3,327 | 4,552 | 6 | 3,703 |
| Pubmed | 19,717 | 44,324 | 3 | 500 |
| Amz-Photo | 7,650 | 287,326 | 8 | 745 |
| Coauthor-CS | 18,333 | 327,576 | 15 | 6,805 |

preserve data sovereignty. The central server aggregates these updates by performing a weighted average proportional to client dataset sizes, refining the global model while reducing communication overhead. Despite its efficiency, FedAvg's convergence guarantees weaken under non-IID data heterogeneity [28], particularly when client distributions exhibit diverging class priors [37].

- **FedDC** [CVPR'22] [15]. A federated learning algorithm that tackles statistical heterogeneity by decoupling and correcting local drift at the client side. Each client learns an auxiliary local drift variable that tracks the parameter gap between its local model and the global model, and then bridges this gap via a penalized consistency term and a gradient correction term in its local objective. By integrating drift correction into the training phase (orthogonal to improved aggregation schemes like FedAdam or FedYogi), FedDC achieves significantly faster convergence and higher accuracy across diverse image-classification benchmarks—including MNIST, Fashion-MNIST, CIFAR-10/100, EMNIST-L, Tiny ImageNet, and a synthetic dataset—while remaining robust under partial participation and large-scale non-IID deployments.

- **FedDyn** [ICLR'21] [2]. A dynamic-regularization–based FL method that resolves the fundamental mismatch between device-level and global optima by augmenting each local objective with a per-round linear and quadratic penalty term, ensuring that, in the limit, local minima coincide with stationary points of the global empirical loss. FedDyn achieves an $\mathcal{O}(1/T)$ convergence rate in both convex and nonconvex settings under partial participation, massive device counts, unbalanced data, and heterogeneity, while requiring only model transmissions (unlike gradient-augmented schemes such as SCAFFOLD). Empirically, it yields substantial communication savings over FedAvg, FedProx, and SCAFFOLD on benchmarks including MNIST, EMNIST-L, CIFAR-10/100, and Shakespeare.

- **FGSSL** [IJCAI'24] [21]. A federated graph learning framework that decouples non-IID hetero-geneity into node-level semantic bias and graph-level structural bias, and corrects both during local training. It introduces Federated Node Semantic Contrast (FNSC), which pulls local node embeddings toward global same-class embeddings and pushes them away from different-class ones, and Federated Graph Structure Distillation (FGSD), which aligns local adjacency-based similarity distributions to those of the global model—all without extra communication rounds or sharing sensitive priors. FGSSL consistently outperforms FedAvg, FedProx, FedOpt, and FedSage on Cora, Citeseer, and Pubmed by up to 4% accuracy under various non-IID settings, while achieving faster, more stable convergence.

- **FedPowder** [ICML'24] [41]. A prompt-based federated continual learning algorithm that fosters dual knowledge transfer along temporal (within-client) and spatial (across-client) dimensions through a two-step prompt aggregation framework: global prompt aggregation via a task corre-lation matrix to capture relevant cross-task information, and top-$k$ correlated prompt selection to optimize communication efficiency; complemented by a correlation-weighted dual distillation loss to preserve transferred knowledge and mitigate catastrophic forgetting. Powder achieves positive forward and backward transfer, substantially lowers communication and storage overhead, and outperforms rehearsal-based, rehearsal-free, and prompt-based baselines on ImageNet-R and DomainNet benchmarks.

- **FedTPP** [NeurIPS'24] [38]. A replay-and-forget-free graph class-incremental learning framework that resolves both inter-task class separation and catastrophic forgetting via two complementary modules: Laplacian smoothing-based task profiling, which yields $100\%$ task ID prediction accuracy by modeling each graph task with a smoothed prototype, and graph prompt learning, which employs a frozen GNN backbone and a small, learnable prompt per task to capture task-specific discriminative information without any data replay. This design achieves zero average forgetting (AF=0) and outperforms state-of-the-art baselines by at least 18% in average accuracy across four large GCIL benchmarks (CoraFull, Arxiv, Reddit, Products).

- **FedSSP**[NeurIPS'24] [50].A federated graph learning algorithm that tackles structural heterogene-ity in cross-domain settings by sharing generic spectral knowledge and adjusting client-specific

preferences. Each client retains non-generic spectral components locally while contributing generic spectral encoders to global collaboration, and employs a personalized preference module with regularization to align extracted features with its own graph structure. By integrating spectral knowledge sharing with preference adjustment—orthogonal to existing federated aggregation schemes—FedSSP achieves superior accuracy and stability across diverse graph-classification benchmarks, including molecular, bioinformatics, social, and vision datasets, while remaining robust under strong non-IID and multi-domain deployments.

- **SEA-ER** [CoLLAs'24] [47].A graph continual learning framework that addresses catastrophic forgetting under structural shifts by analyzing the theoretical learnability of GNNs and introducing a structure-aware replay mechanism. The paper first proves that GNNs may become unlearnable in node-wise continual learning when evolving graph structures induce large distributional shifts, highlighting the central role of topological integrity. To mitigate this, it proposes Structure-Evolution-Aware Experience Replay (SEA-ER), which selects representative samples via topology-aware structural similarity and reweights them through structural alignment in the replay phase. By integrating these strategies into the training process (orthogonal to architectural advances in GCN, GAT, or SAGE), SEA-ER achieves markedly higher stability and accuracy across real-world and synthetic benchmarks—including OGB-Arxiv, Reddit, and CoraFull—while remaining effective under evolving graph dynamics and large-scale non-IID continual learning deployments.

### C.3 Ablation study of conflicts in parameter aggregation.

We conducted a systematic evaluation of parameter similarity across clients by measuring the cosine similarity of their update vectors on the Cora dataset. As shown in Table 4, the results indicate substantial divergence among client parameters, reflected in both their spatial distribution and update directions. This confirms the presence of parameter conflicts caused by statistical heterogeneity during federated aggregation. From the perspective of parameter similarity, these findings motivate the integration of the G-EPAE module on the server side. By constructing a graph-structured compatibility matrix, G-EPAE adaptively identifies and reconciles discrepancies in client updates, thereby reducing aggregation conflicts and improving both the generalization ability and training stability of the global model.

Table 4: **Ablation study on the Cora dataset of conflicts in parameter aggregation.**

| Cos similarity | Client1 | Client2 | Client3 | Client4 | Client5 |
|---|---|---|---|---|---|
| Client1 | 1.00 | 0.63 | 0.57 | 0.55 | 0.61 |
| Client2 | 0.63 | 1.00 | 0.56 | 0.56 | 0.65 |
| Client3 | 0.57 | 0.56 | 1.00 | 0.60 | 0.54 |
| Client4 | 0.55 | 0.56 | 0.60 | 1.00 | 0.50 |
| Client5 | 0.61 | 0.65 | 0.54 | 0.50 | 1.00 |

### C.4 Discussion of long-range temporal retention.

In the context of FCGL, preserving long-range knowledge is critically important. To evaluate the ability of the model to retain early-task information over extended sequences, we tracked task-specific classification accuracy at each time step. As shown in Table 5, the accuracy trajectories demonstrate that performance on early tasks exhibits only negligible degradation as the task sequence grows, and in some cases even improves. This suggests that sequence length does not impose a substantial negative impact on the accuracy of MOTION and, in fact, reveals a trend of sustained optimization for certain tasks. We attribute this stability and improvement mainly to the design of the G-TMSC module. By employing a multi-expert, topology-aware coarsening strategy, the module preserves and amplifies critical structural information from early tasks within the compressed graphs. As a result, these topological signals are consistently retained and reinforced during subsequent incremental updates, which not only mitigates knowledge drift but also enables, to some extent, the positive consolidation of prior knowledge.

### C.5 Discussion of compatibility with GNN architectures

MOTION is model-agnostic, as it operates at both the parameter update level and the structural summary level. This design allows seamless integration with various GNN architectures, such as

Table 5: **Accuracy evolution of tasks over time steps on Cora dataset.**

| Accuracy | Task1 | Task2 | Task3 | Task4 | Task5 | Task6 | Task7 |
|----------|-------|-------|-------|-------|-------|-------|-------|
| TimeStep1 | 0.70 | 0.00 | 0.00 | 0.00 | 0.00 | 0.00 | 0.00 |
| TimeStep2 | 0.72 | 0.66 | 0.00 | 0.00 | 0.00 | 0.00 | 0.00 |
| TimeStep3 | 0.75 | 0.68 | 0.36 | 0.00 | 0.00 | 0.00 | 0.00 |
| TimeStep4 | 0.74 | 0.86 | 0.37 | 0.51 | 0.00 | 0.00 | 0.00 |
| TimeStep5 | 0.85 | 0.84 | 0.42 | 0.64 | 0.54 | 0.00 | 0.00 |
| TimeStep6 | 0.82 | 0.81 | 0.57 | 0.66 | 0.55 | 0.53 | 0.00 |
| TimeStep7 | 0.82 | 0.79 | 0.58 | 0.65 | 0.57 | 0.54 | 0.60 |

GCN and GAT. In our experiments 1, we primarily adopted GAT for consistency with baselines. However, as shown in Table 6, we also evaluated MOTION with GCN on three datasets, where it achieved comparable performance gains, further confirming its broad compatibility across GNN architectures.

Table 6: **Compatibility of MOTION with GNN architectures.**

| Method | Cora | | CiteSeer | | PubMed | |
|--------|------|------|----------|------|--------|------|
| | AA | AF | AA | AF | AA | AF |
| FedPowde[NeurIPS24] | 0.14 | 0.73 | 0.15 | 0.56 | 0.41 | 0.30 |
| FedTpp[ICML24] | 0.26 | 0.27 | 0.17 | 0.24 | 0.39 | 0.10 |
| MOTION | **0.63** | **-0.09** | **0.52** | **-0.09** | **0.61** | **-0.13** |

## C.6 Implementation Details.

All experiments were conducted on a system equipped with an NVIDIA GeForce RTX 3090 GPU (24 GB), an Intel Xeon Gold 6330 CPU @ 2.00 GHz (14 cores, 28 threads), and 90 GB of RAM, running Ubuntu 22.04 with Python 3.12, PyTorch 2.3.0, and CUDA 12.1. We employed the Graph Attention Network (GAT) as our base model and evaluated federated graph learning on five benchmark datasets: Cora, CiteSeer, PubMed, Amazon-Photo, and CoAuthor-CS. The federated setup comprised two clients participating in a single communication round using the FedAvg algorithm. To induce label skew, we partitioned each dataset's eight classes across clients according to a Dirichlet distribution with concentration parameter $\alpha = 5.0$. Our GAT architecture consisted of three attention layers with a hidden dimension of 64, a dropout rate of 0.3, a learning rate of 0.005, and a weight decay of $4 \times 10^{-4}$. We split each dataset into 20% training, 40% validation, and 40% test subsets, fixed the random seed at 4 for reproducibility, and performed all computations on GPU 0.

## D Broader impact

Our work represents a pioneering step towards a generalizable FCGL framework that simultaneously preserves graph-topological integrity and mitigates parameter conflicts in decentralized, evolving environments. By introducing the `MOTION` framework, which integrates G-TMSC and G-EPAE, we enable robust incremental learning across dynamic, distributed graph data. This advancement enhances the generalization performance and stability of graph neural networks in the FCGL setting and contributes to the broader objective of developing scalable, privacy-preserving, lifelong graph intelligence systems.

## E Limition

Despite the demonstrated effectiveness of `MOTION` in addressing the challenges of FCGL, this framework has certain limitations. The inherent tension between preserving graph–topological information and ensuring efficient aggregation under extreme client heterogeneity remains an open research problem within FCGL scenarios. Moreover, our evaluation, although conducted on diverse benchmark datasets, takes place in controlled settings that may not capture the full complexity of real-world federated systems, including severe client heterogeneity, unstable participation, and adversarial behaviors. Future work could investigate more adaptive mechanisms for handling abrupt

topological changes, develop defense strategies against malicious clients, and validate `MOTION` in realistic deployment environments.

