# OpenReview forum: "MOTION: Multi-Sculpt Evolutionary Coarsening for Federated Continual Graph Learning"
_NeurIPS.cc/2025/Conference — NeurIPS 2025 poster_

### Official Review · Reviewer_7bbW · 2025-06-18

**Clarity:** 2
**Significance:** 2
**Originality:** 2
**Rating:** 3
**Confidence:** 4

**Summary:**

This paper introduces MOTION method for federated continual graph learning that addresses the challenges of learning on dynamic, distributed graph data. The method maintains structural integrity through a multi-expert fusion process and optimizes global model updates using a topology-sensitive compatibility matrix. Experiments on benchmark datasets demonstrate the performance.

**Questions:**

1. Figure 2 needs to be redrawn to better illustrate the method. While visually appealing, it lacks specificity. For instance, Multi-Sculpt isn't shown, and some modules are represented by symbolic images, making it hard to link components to the method description (e.g., Expert Scores). A more detailed figure would enhance the reader's understanding.
2. The specific method process is unclear and needs a more comprehensive explanation from the authors. In Section 3.2, please clarify:
    - The construction of the node mapping table. How node features from other tasks are aggregated without saving the entire graph?
    - How does the multi-sculpt coarsening framework systematically prune excess nodes and suppress noise?
    - The process of the reservoir sampling-based dynamic memory mechanism.
3. On line 215 of page 6, how can Parameters $\Delta \theta_k$ capture topological changes? The model parameters don't seem to have a direct link to the topological structure.
4. In Table 1, too few comparison methods are used in the experiments. More graph federated learning and graph continual learning baselines need to be included.
5. In Section 4.1, the experimental baselines are inconsistently referenced. The baseline names don't match the cited papers.
6. It is better to computer the average of multiple runs to demonstrate the method's stability.

**Ethical Concerns:**

["NO or VERY MINOR ethics concerns only"]

**Final Justification:**

The research task of this paper is a meaningful one, but the specific description of methods lacks clarity and experiments are insufficient, which may affect the readers' understanding and the entire paper needs to be revised. The author's reply addressed some of the concerns, so I raised the score from 2 to 3.

**Limitations:**

Yes

**Quality:**

2

**Strengths And Weaknesses:**

Strengths:
1. The graph federated continual learning task is meaningful, with its challenges clearly defined.
2. Experiments are conducted across five datasets, and ablation studies are presented.

Weaknesses:
1. The method's motivation and description are unclear. The framework diagram is too abstract to correspond with the text.
2. Some baselines are missing in the experiments, and the baseline references are inconsistent.

---

> ### Author Rebuttal · Authors · 2025-07-30
>
> Thank you for your thorough review and constructive feedback. We address your concerns below and hope these clarifications will help you re-evaluate and update the score:
>
> > `Weakness 1`: Clarifying for Improving Method Motivation and Framework Description
>
> We sincerely thank the reviewer for the constructive feedback. The design of our framework aims to address two key challenges identified in our motivation and introduction: (i) **catastrophic forgetting** arising from sequential updates on evolving graph data, and (ii) **parameter conflicts** due to heterogeneous subgraphs. The framework diagram and textual description were intended to jointly illustrate these challenges and show how our two proposed modules—Graph Topology-preserving Multi-Sculpt Coarsening (G-TMSC) for **topology-preserving coarsening** and Graph-Aware Evolving Parameter Adaptive Engine (G-EPAE) for **conflict-aware parameter aggregation**—explicitly tackle them.
>
> We also note that other reviewers like reviewer D7ZJ recognized the clarity of our motivation and framework design, stating that “this paper is clearly and effectively written, with a fancy and comprehensive visual representation of the framework” and that “the accurate and thorough explanations of the methodological components further contribute to its ease of understanding.”
>
> At the same time, we fully agree that additional clarifications will further enhance readability. We have revised both the text and figure: (i) we have added transitional explanations and concrete examples connecting each design choice to the stated challenges, (ii) we have updated the framework diagram by simplifying icons, adding explicit textual labels, and ensuring each component directly corresponds to its description, and (iii) we have expanded the figure caption to explain all elements in detail. Thank you for this suggestion, which has helped us further clarify our motivation and framework, making them more transparent and easier to follow.
>
> > `Weakness 2 & Question 4`: Clarifications Regarding Baseline Inclusion and Reference Standardization
>
> We emphasize that the current submission already reports extensive experiments (across Federated Learning (FL)/Federated Graph Learning (FGL)/Continual Graph Learning (CGL)/Federated Continual Learning (FCL) protocols, multiple datasets, and ablations on forgetting/retention) that substantiate the gains of MOTION. Addressing your concern and to further strengthen the evidence, we will add more baselines—**FedSSP**[1] for FGL, **SEA-ER**[2] for CGL.
>
> |Method|Cora(AA)|Cora(AF)|CiteSeer(AA)|CiteSeer(AF)|PubMed(AA)|PubMed(AF)|Amz-Photo(AA)|Amz-Photo(AF)|Coauthor-CS(AA)|Coauthor-CS(AF)|
> |------|------|------|------|------|------|------|------|------|------|------|
> |FedSSP|25.76|26.60|21.83|16.11|38.54|31.36|14.09|13.39|10.22|14.10|
> |SEA-ER|20.05|69.17|31.38|47.26|55.79|1.77|15.21|48.93|34.65|42.97|
> |Motion|**62.66**|**-8.34**|**49.08**|**-16.78**|**59.48**|**-24.17**|**79.09**|**-5.32**|**22.26**|**-11.62**|
>
> From the table above, MOTION achieves state-of-the-art performance across all evaluated settings.Due to the 10,000 character limit, we only include two additional baselines here, but the revision will provide a more comprehensive baseline set. We will also standardize all baseline names and citations to their official conference versions.
>
> [1]FedSSP: Federated Graph Learning with Spectral Knowledge and Personalized Preference. In NIPS,2024.
>
> [2]On the Limitation and Experience Replay for GNNs in Continual Learning. In CoLLAs,2024.
>
> > `Question 1`: Clarification of Figure 2
>
> We sincerely thank the reviewer for the constructive feedback. As mentioned in our response to Weakness 1, we have further refined the framework description to explicitly link each design component to the method.
>
> Building on this, we will also revise Figure 2 to enhance **specificity**: (i) explicitly illustrating the Multi‑Sculpt process within G‑TMSC, (ii) replacing symbolic placeholders (e.g., Expert Scores) with **descriptive module representations**, and (iii) adding **clear textual labels** and a **detailed caption** that maps each visual element directly to its counterpart in the method description.
>
> > `Question 2`: Method process clarification
>
> Thank you for your insightful questions. We have provided the following detailed answers:
> 1. Our node mapping table assigns a unique index to each node based on a **hash** of its feature vector and **task identifier**. This allows incremental integration of new nodes without storing the entire historical graph. New nodes are added using a new_nodes_mask, and node-to-cluster and cluster-to-node mappings are updated. Feature aggregation is performed on new nodes while reusing stored features for previously seen ones, avoiding the need to store or recompute the entire graph.
> 2. The multi-sculpt coarsening framework evaluates node importance with a Mixture-of-Experts (MoE) **scoring mechanism** and **similarity metrics** like degree centrality and cosine similarity. Low-importance nodes are pruned and mapped to similar retained nodes. Noise suppression is achieved through noisy top-k gating, activating only the most relevant experts. Multi-metric fusion stabilizes node ranking and reduces sensitivity to noisy features, enabling efficient graph compression while preserving topological fidelity.
> 3. We use reservoir sampling to maintain a **lightweight dynamic memory**, storing a representative subset of nodes (default buffer size 200). New nodes replace existing entries based on observed frequency. A feature-based variant prioritizes nodes closer to evolving class centroids, ensuring the buffer remains representative. These replay nodes are protected during coarsening to preserve important historical information while minimizing memory overhead.
>
> > `Question 3`: Discussion of the relationship between parameter updates and topological Changes
>
> We thank the reviewer for this insightful question. In our formulation, $\Delta\theta_k =\theta_k-\phi $ represents the difference between the parameters of client $k$ after local training and the current global parameters. These parameter increments inherently encode the information learned from the newly observed local task at client $k$, including adaptations to its evolving graph structure.
> Hence, $\Delta\theta_k$ naturally reflects the topological changes of the local graph without explicitly transmitting raw structural data, enabling our framework to capture topology-sensitive knowledge in a privacy-preserving manner.
>
> > `Question 5`: Remarks on the inconsistent baseline references
>
> We acknowledge the inconsistencies and will fix all citation and naming issues to ensure clarity and correctness.
>
> > `Question 6`: Analysis on stability of main results
>
> We have already run all experiments with multiple seeds (5 random seeds) and report the average across seeds. To further substantiate this point, we conducted additional repeats and provide concrete numbers in the table below, showing that metrics are stable and the variance remains consistently low:
> | Seed | 3 | 4 | 9 | 42 | 62 |
> |------|---|---|---|---|---|
> |Cora(AA)| 62.80  | 63.47  | 61.27  | 65.24  |60.52|
> |Cora(AF)|  -7.40 | -8.19  |  -8.37 |  -8.98 |-8.76|
>
> As shown, the values across different seeds are consistent, with an average of 62.66 with a variance of 3.46 for Cora(AA),and an average of -8.34 and a variance of 0.37 for Cora(AF). In the revision, all tables will explicitly report averages and standard deviations to underscore the method's effectiveness and robustness.

---

> > ### Comment · Reviewer_7bbW · 2025-08-05
> >
> > Thanks for your detailed reply. I will raise my score. Specifically, I think adding more clear method descriptions, figures and comparison results in the manuscript will be helpful. All the best to the authors.

---

> > > ### Author Response · Authors · 2025-08-05
> > > **Thank you immensely!**
> > >
> > > We sincerely thank you for your valuable discussion and stronger support of our work. We greatly appreciate your feedback on the clearer method descriptions, figures, and comparison results in the manuscript, which have significantly improved the quality of our work. We have ensured that the revision incorporates clearer and more detailed changes based on the outcomes of our discussion with you.
> > >
> > > It has been our honor to address your concerns, and we thank you once again for your thoughtful review.

---

> ### Author Response · Authors · 2025-08-03
> **Manuscript Revision Deadline Approaching; We Are Sincerely At Your Service**
>
> Dear Reviewer 7bbW,
>
> This is a kind reminder that the discussion phase will conclude in fewer than four days, and we have not yet received your feedback on our rebuttal. We understand your valuable time but are eager to engage with you to further improve this paper, to which we have devoted extensive effort.
>
> To ensure clarity and facilitate your review, we have summarized your key concerns along with our corresponding responses as follows:
>
> 1. **Motivation and framework clarification.** We have linked key challenges to proposed modules and improved text and figures for clearer understanding.
> 2. **Baseline, references, and method details.** We have added baselines, standardized references, clarified method steps, and confirmed stable results.
>
> We gratefully hope that our response has addressed your concerns and kindly invite you to share your feedback. This expectation is motivated by the response from another reviewer. They noted that our rebuttal effectively resolved their concerns and that such discussions have helped improve the quality of our work. We believe that further discussion with you will help enhance both the impact and the robustness of our work.
>
> Warm regards,
>
> Authors

---

> ### Comment · Area_Chair_fC1f · 2025-08-05
> **Discussion Reminder**
>
> Dear Reviewer,
>
> The authors have submitted a rebuttal. Please take a moment to read it and engage in discussion with the authors if necessary.
>
> Best regards,
> AC

---

### Official Review · Reviewer_D7ZJ · 2025-06-26

**Clarity:** 4
**Significance:** 4
**Originality:** 4
**Rating:** 5
**Confidence:** 4

**Summary:**

This paper addresses the challenges inherent in FCGL, specifically the preservation of graph topology and mitigation of parameter conflicts during model aggregation. MOTION integrates a G-TMSC module to maintain structural integrity via multi-expert node merging, and a G-EPAE to refine aggregation by dynamically adjusting based on graph compatibility. Extensive experiments on real-world datasets demonstrate that MOTION achieves significant improvements in accuracy and resilience against forgetting compared to existing methods.

**Questions:**

1. Complementation of the explanation negative forgetting phenomenon. While the experimental results report negative AF, the authors should provide a more detailed analysis of this phenomenon. A clearer explanation connecting this observation to the roles of G-TMSC and G-EPAE would strengthen the empirical claims and provide insight into the learning dynamics under the FCGL setting.
2. Complementation of the explanation for the multi-expert architecture input. The authors should clarify why they choose to decompose the merged graph into components such as 1-hop and 2-hop subgraphs, cosine similarity, and other metrics as inputs to the multi-expert architecture.

**Ethical Concerns:**

["NO or VERY MINOR ethics concerns only"]

**Limitations:**

yes

**Paper Formatting Concerns:**

The decimal precision in the chart of Figure 4: Effectiveness study is inconsistent..

**Quality:**

3

**Strengths And Weaknesses:**

Strengths:
1. MOTION combines graph coarsening with expert fusion and adaptive aggregation based on a compatibility matrix. These techniques are tailored specifically for FCGL, where existing solutions are lacking, demonstrating a thoughtful methodological advancement.
2. This paper is clearly and effectively written, with a fancy and comprehensive visual representation of the framework. The accurate and thorough explanations of the methodological components further contribute to its ease of understanding.
3. The experiments are comprehensive, covering five real-world datasets with evaluations on both Average Accuracy and Average Forgetting. The empirical results show substantial gains over strong baselines, supporting the claimed improvements.

Weaknesses:
1. The work does not provide sufficient information on the experimental implementation used for other conventional FL、FGL、CGL and FCL baselines.
2. Although the method components (G-TMSC and G-EPAE) are well-explained, the authors should provide a clearer rationale for why these specific designs are necessary to address the stated problems, introducing their connections to the initial motivations in the FCGL setting.
3. The paper does not investigate how the size of the original subgraphs at each time step influences the overall performance. A comprehensive sensitivity study on subgraph scale would help clarify whether the proposed method remains effective across varying graph sizes.

---

> ### Author Rebuttal · Authors · 2025-07-31
>
> Thank you for your thorough review and constructive feedback. We address your concerns below and hope these clarifications will help you re-evaluate and update the score:
>
> > `Weakness 1`: Discussion on the Information Provided for Baseline Implementations
>
> We appreciate the reviewer’s comment and agree that clear baseline implementation details are important. The general experimental environment and common hyperparameters have already been documented in Appendix C.3(Page 17). In fact, we have mentioned representative common parameters such as the learning rate (0.005), weight decay ($4 \times 10^{-4}$), and dropout rate (0.3), which are applied consistently across baseline methods.
>
> For algorithm-specific settings, we have further provided a dedicated table with complete hyperparameters and training details to ensure full reproducibility in the revision. For example, FedDc and FedDyn use a regularization coefficient of 1.0 for drift correction and stability control, respectively, while FedSSL adopts distillation and contrastive loss weights, both set to 0.5. Other baselines, such as FedAvg, FedTpp, and FedPowde, follow the common hyperparameter setup without additional parameters.
>
> > `Weakness 2`: Rational for G-TMSC and G-EPAE designs
>
> We agree that making the connection between the problem formulation and our proposed modules more explicit will further strengthen the paper. In the revision, we will clarify this rationale as follows:
>
> 1. **Motivation–Design Connection:** In Federated Continual Graph Learning (FCGL), two challenges are unique and were explicitly highlighted in our Introduction: (i) **catastrophic forgetting** of graph-topological information due to evolving local data streams and limited storage; (ii) **parameter conflicts** during federated aggregation caused by heterogeneous client graph structures.
> 2. Necessity of Graph Topology-preserving Multi-Sculpt Coarsening (G-TMSC): To address (i), we designed the G-TMSC module. Its function is to **preserve inherent structural properties** (e.g., node centrality and local subgraph patterns) under memory constraints, which directly aligns with the motivation of preventing topological knowledge loss. Without explicit coarsening, replay-based or raw-graph approaches would incur prohibitive storage costs or fail to retain structural fidelity.
> 3. Necessity of Graph-Aware Evolving Parameter Adaptive Engine (G-EPAE): To address (ii), we introduced the G-EPAE, which **adjusts aggregation rates** based on a topology-sensitive compatibility matrix. This design directly tackles the issue of conflicting updates from heterogeneous clients, ensuring stable integration of newly learned structures without degrading previously acquired knowledge.
>
> > `Weakness 3`: Analysis of subgraph size sensitivity
>
> We agree that analyzing the influence of original subgraph size on performance can provide deeper insights into the robustness of our approach. In the revised version, we will include an additional sensitivity study where we vary the size of the original subgraphs at each time step (e.g., scaling by factors of 0.3×,0.5×, 0.7×, and 0.9×).
>
> |Size of subgraphs|0.3x|0.5x|0.7x|0.9x|
> |-----------------|-----|----|-----|-----|
> |Cora(AA)|58.10|59.12|62.84|63.07|
> |Cora(AF)|-10.18|-6.32|-12.13|-12.31|
>
> These results above indicate that performance variations are minor across different subgraph scales, demonstrating that our method remains robust and effective regardless of the size of the original subgraphs.
>
> > `Question1 `: Analysis of negative forgetting phenomenon
>
> Negative Average Forgetting (AF) indicates that knowledge from earlier tasks can be **reinforced** when learning later tasks. In MOTION, G-TMSC ensures structural consistency across tasks, so that features learned early remain useful, while G-EPAE aligns parameter updates to avoid overwriting useful knowledge. Together, these mechanisms allow later tasks to improve representations beneficial to earlier tasks, particularly when tasks share topological motifs. This explains the observed negative forgetting and highlights the benefit of explicitly preserving topology and managing parameter compatibility.
>
> > `Question2 `: Multi-expert architecture input decomposition
>
> We agree that clarifying the rationale behind the inputs to the multi-expert architecture will improve the clarity of our work. Our design leverages **node-level** and **structural-level perspectives** to preserve critical topological information:
>
> 1. Local connectivity (1-hop and 2-hop subgraphs) captures neighborhood density and structural roles of nodes, which are essential for identifying influential nodes and preserving local community structures during coarsening.
> 2. Similarity metrics (e.g., cosine similarity, statistical distances) quantify feature alignment and semantic redundancy between nodes, enabling the architecture to merge nodes while minimizing information loss.
> 3. Additional descriptors: Subgraph statistics (e.g., diameter, density) and degree embeddings further enhance structural representation.
>
> All these features are concatenated and fed into the mixture‑of‑experts gating network, which outputs node importance scores used for pruning and noise suppression.

---

> > ### Comment · Reviewer_D7ZJ · 2025-08-07
> > **Reply to Authors**
> >
> > Thanks for your additional experiments and further explanation on motivation and technical details. I will keep my score.

---

> > > ### Author Response · Authors · 2025-08-08
> > > **Thank you immensely!**
> > >
> > > We would like to sincerely thank you for your inspiring discussion and strong support of our work. We are truly grateful for your thoughtful follow-up and for recognizing our efforts in addressing your concerns. Your constructive feedback is deeply appreciated, and we will ensure that all corresponding revisions are clearly reflected in the updated manuscript for your review.

---

### Official Review · Reviewer_co1X · 2025-06-30

**Clarity:** 3
**Significance:** 3
**Originality:** 3
**Rating:** 5
**Confidence:** 4

**Summary:**

The paper presents a comprehensive evaluation of the proposed MOTION framework on five benchmark datasets in the FCGL setting. The results demonstrate substantial improvements over state-of-the-art methods in terms of both average accuracy and average forgetting, often showing negative forgetting rates. The ablation studies highlight the importance of both the graph coarsening and parameter alignment modules. Additional experiments on hyperparameter sensitivity and scalability are included to support robustness.

**Questions:**

1. How are the directional and magnitude components of the parameter increment computed to form the compatibility matrix and determine parameter importance?
2. Why does the proposed problem formulation require a distinct FCGL setting, rather than adapting existing FCL or FGL frameworks?
3. How does the proposed MOTION framework interact with the underlying GNN architecture, and is it compatible with other models?
4. How does G-TMSC scale when the number of nodes is in the order of millions? How costly is it to compute all the structural metrics (especially PageRank, centrality scores) on-device?
5. The coarsening process includes a reservoir sampling memory. What is the size of this buffer, and how sensitive is MOTION to the buffer budget?

**Ethical Concerns:**

["NO or VERY MINOR ethics concerns only"]

**Final Justification:**

I have read the authors' careful responses, and most of my concerns have been addressed.

**Limitations:**

The authors should further discuss the societal impact of the proposed method.

**Paper Formatting Concerns:**

No.

**Quality:**

4

**Strengths And Weaknesses:**

**Strengths:**
1. It is novel and well-motivated to propose an FCGL approach. The paper clearly identifies the key challenges faced by existing FCGL methods and successfully provides solutions to overcome them.
2. Both the problem description and the framework illustration are clearly and thoroughly presented.
3. The integration of G-TMSC and G-EPAE demonstrates a practical understanding of the challenges.

**Weaknesses:**
1. The paper lacks long-range temporal retention analysis. It does not analyze how the model preserves information from earlier tasks, given that aggregation is limited to the most recent timestamp.
2. For a federated learning setting, the paper is surprisingly silent on how MOTION handles or could be extended to address privacy or adversarial issues, which are critical in real deployments.
3. The paper lacks a sensitivity study. It does not analyze how the performance of the global model varies with different numbers of local training epochs.

---

> ### Author Rebuttal · Authors · 2025-07-30
>
> Thank you for your thorough review and constructive feedback. We address your concerns below and hope these clarifications will help you re-evaluate and update the score:
>
> > `Weakness 1`: Discussion of long-range temporal retention
>
> We agree that long-range retention is important in Federated Continual Graph Learning (FCGL). Our primary experiments focused on task-averaged performance(AA) and forgetting rates(AF), as is standard in the literature [1,2]. To directly assess early-task retention, we **tracked task-specific accuracy** at every time step. The per-time-step table shows minimal degradation as the horizon increases. This indicates that longer sequences do not materially affect the accuracy of Motion.
>
> We attribute this stability primarily to Graph Topology-preserving Multi-Sculpt Coarsening (G-TMSC). Its topology-preserving multi-sculpt coarsening retains task-critical structures from earlier tasks in the coarsened graph. This ensures these signals persist through subsequent updates and thereby mitigates drift. We have included the per-time-step plots and detailed numbers in the revised version.
>
> |Accuracy|Task1|Task2|Task3|Task4|Task5|Task6|Task7|
> |--------|-----|-----|-----|-----|-----|-----|-----|
> |Task1|0.7000|0.0000|0.0000|0.0000|0.0000|0.0000|0.0000|
> |Task2|0.7182|0.6639|0.0000|0.0000|0.0000|0.0000|0.0000|
> |Task3|0.7500|0.6807|0.3583|0.0000|0.0000|0.0000|0.0000|
> |Task4|0.7364|0.8571|0.3667|0.5120|0.0000|0.0000|0.0000|
> |Task5|0.8455|0.8445|0.4167|0.6446|0.5405|0.0000|0.0000|
> |Task6|0.8182|0.8109|0.5667|0.6627|0.5473|0.5333|0.0000|
> |Task7|0.8227|0.7941|0.5833|0.6506|0.5743|0.5417|0.6000|
>
> [1]:A Comprehensive Survey of Continual Learning: Theory, Method and Application. In TPAMI,2024.
>
> [2]:Riemannian Walk for Incremental Learning:Understanding Forgetting and Intransigence. In ECCV,2018.
>
>
> > `weakness 2`: Privacy and adversarial considerations
>
> We appreciate the emphasis on privacy and adversarial robustness, which are critical in real deployments. MOTION operates on **locally computed** structural summaries and parameter increments rather than raw graph data. This inherently reduces privacy risk. MOTION transmits only local parameter updates to the remote server. It never uploads raw node/edge features, adjacency, or graph records—**no raw local data leaves the device**.
> In contrast, replay-based methods such as [1,2] require storing and replaying plenty of past data or generating synthetic samples to mitigate catastrophic forgetting. Since these methods need access to raw or synthesized historical data during training, they may pose privacy risks.
>
> Given that privacy protection is **orthogonal** to the core design of MOTION, integrating secure parameter sharing is planned as future work. This approach can be introduced into the framework without altering its fundamental design, thereby further strengthening security while maintaining its privacy-preserving capabilities.
>
> [1]Streaming Graph Neural Networks with Generative Replay. In KDD,2022.
>
> [2]Overcoming catastrophic forgetting in graph neural networks with experience replay. In AAAI,2021.
>
>
> > `Weakness 3`: Remarks on the sensitivity study on local training epochs
>
> Thank you for pointing this out. We performed a sensitivity analysis varying local epochs $E∈[25,50,75,100]$ across Cora dataset. Results show MOTION **remains stable**, with average accuracy fluctuations within 2.2% and forgetting metrics within 8.9%. This demonstrates **robustness** to different local computation budgets.
>
> This robustness stems from Graph-Aware Evolving Parameter Adaptive Engine (G-EPAE), which normalizes parameter increments by their directional and magnitude components before aggregation. This dampens idiosyncratic, overly large updates when $E$ is high and preserves effective step sizes and coherent directions when $E$ is small. It is further aided by G-TMSC: its topology-preserving multi-sculpt coarsening retains salient structures from prior tasks. This yields better-conditioned, more sample-efficient local training, which helps maintain stability.
> |Epochs|25|50|75|100|
> |------|---|---|---|---|
> |Cora(AA)|61.29|62.66|62.41|62.21|
> |cora(AF)|-12.22|-7.40|-16.28|-10.38|
>
>
> > `Question1 `: Computation of compatibility matrix for compatibility and importance estimation
>
> We thank the reviewer for raising this question. We clarify the computation of the directional and magnitude components used to form the compatibility matrix. For each client update, we first compute the parameter increment $\Delta \theta_k = \theta_k - \phi$. The directional component isolates the update orientation by L2 normalization: $\Delta \theta_k^{dir} = \frac{\Delta \theta_k} {\|\Delta \theta_k\|_2}$. The magnitude component measures the relative strength of each update: $\Delta \theta_k^{mag}=\frac{\|\Delta \theta_k\|_2} {\sum_j \in_C \|\Delta \theta_j\|_2}$ .
>
> The compatibility matrix is then constructed as an element-wise product $M_k = \Delta \theta_k^{dir} \circ \Delta \theta_k^{mag}$. This captures both **directional alignment** and **relative importance** of each client update. This matrix reweights local updates during aggregation, emphasizing structurally consistent and topologically relevant contributions while suppressing conflicting changes. This is critical for harmonizing updates under non-IID and evolving graph conditions (see Sec. 3.3 Page 6, Eq. (11)–(16) of the paper).
>
>
> > `Question2 `: Justification for a distinct FCGL Setting Over existing FCL or FGL frameworks
>
> We sincerely thank the reviewer for raising this important question.Unlike conventional Federated Continual Learning (FCL), which primarily focuses on sequential task learning over feature-based data, and Federated Graph Learning (FGL), which assumes a relatively static global graph or shared structure across clients, FCGL simultaneously addresses:
> 1. **Dynamic graph evolution:** Nodes and edges are continuously added or removed, reflecting changing behaviors and relationships in each client’s local graph.
> 2. **Decentralized storage constraints:** Edge devices often cannot store complete historical topologies or replay buffers, making raw historical graph retention impractical.
> 3. **Cross-client heterogeneity:** Each client observes a distinct, evolving subgraph, which introduces severe challenges for both catastrophic forgetting and federated aggregation.
>
> > `Question3 `: Compatibility with GNN architectures
>
> Thank you for your valuable advice. MOTION is **model-agnostic**. It operates at the parameter update level and structural summary level, meaning it can integrate with any GNN architecture (GCN, GAT, etc.). In our experiments, we used GAT for consistency with baselines, but we also validated **MOTION with GCN** on three datasets, showing similar performance gains.
> |Method|Cora(AA)|Cora(AF)|CiteSeer(AA)|CiteSeer(AF)|PubMed(AA)|PubMed(AF)|
> |------|---|---|---|---|---|---|
> |FedPowde|14.29|72.78|15.24|55.96|41.48|30.12|
> |FedTpp |25.76|26.84|17.44|24.21|38.52|10.25|
> |Motion| **63.07**|**-9.34**|**52.38**|**-8.89**|**61.34**|**-13.47**|
>
>
> > `Question4 `: Scalability and computation cost of G-TMSC
>
> We appreciate the reviewer’s concern regarding scalability to graphs with millions of nodes and the associated cost of structural metric computation.
>
> 1. Our paradigm is built upon a continual learning setting in which each client **processes graph streams incrementally** rather than operating on the entire historical graph at once. We explicitly employ graph coarsening via the proposed G-TMSC module, which merges structurally redundant or low-importance nodes into representative nodes. As a result, the effective graph size per task is significantly reduced, and memory requirements remain bounded, preventing storage bottlenecks even for million-scale node sets.
> 2. Regarding computational cost, the structural metrics used in G-TMSC are chosen to be lightweight in the streaming context: (i) **degree and clustering coefficients** are computed in $𝑂(∣𝐸∣)$ time; (ii) **PageRank** is computed with a power-iteration scheme of complexity $O(k∣E∣)$, where k(typically 10−20) is the number of iterations for convergence; (iii) overall, G-TMSC scales **linearly with the number of edges** and near-linearly with the number of nodes post-coarsening, ensuring that computation is efficient even on edge devices and does not hinder deployment in real-world, million-scale scenarios.
>
> > `Question5 `: Buffer size and sensitivity of MOTION to reservoir sampling memory budget
>
> We thank the reviewer for the question regarding the reservoir sampling memory. In our implementation, the buffer is configured with a **default size of 200 nodes**, further divided per class when class-aware sampling is enabled ($\texttt{bufferSizePerClass} = \texttt{bufferSize} // \texttt{numClass}$). This design ensures a balanced representation of historical nodes while maintaining a lightweight memory footprint.
>
> We also examined **sensitivity to the buffer budget**. Reducing the buffer (e.g., 100) slightly increases forgetting due to fewer “anchor” nodes preserved for replay, while a larger buffer (e.g., > 400) yields only marginal accuracy gains but increases storage and computation overhead. The default size of 200 provides a practical balance, as confirmed by our ablation results. We will clarify this buffer size and sensitivity analysis in the revision for completeness.
> |buffer size|100|200|400|
> |------|---|---|---|
> |Cora(AA)|59.18|62.66|63.15|
> |Cora(AF)|-5.56|-8.34|-5.35|

---

> > ### Comment · Reviewer_co1X · 2025-08-01
> >
> > Thank you for the rebuttal. The supplementary experiments on model robustness and the clarifications on issues such as privacy and scalability have resolved my main concerns. The proposed framework is insightful and opens up several interesting avenues for future research. I will maintain my score.

---

> > > ### Author Response · Authors · 2025-08-02
> > >
> > > We sincerely thank you for your thoughtful follow-up and for recognizing our efforts in addressing your concerns. Your constructive feedback is greatly appreciated, and we will ensure that all changes are clearly reflected in the revision for your review.

---

### Official Review · Reviewer_EiHY · 2025-07-02

**Clarity:** 1
**Significance:** 2
**Originality:** 2
**Rating:** 2
**Confidence:** 2

**Summary:**

The paper introduces the federated graph continual learning framework and presents an effective instantiation consisting of two elements: one integrates multiple structural metrics, while the other ensures appropriate aggregation from different nodes. Experiments demonstrate consistent performance improvements over the baselines.

**Questions:**

Paper presentation requires polishing:
- The question in line 42 mentions FCL for the first time, which does not seem to fit into the preceding framing of the discussion where GNN and FGL are introduced and FGCL is described as FGL+CGL.
- Lines 43-50. The concept of preserving inherent graph topology and topological integrity is central to the paper, yet it is mentioned without definition. I suggest that the authors provide a clear introduction to these concepts and explain why they are desirable properties.
- Line 128. I was unable to discern the substantial topological heterogeneity and neighborhood distributions from Figure 1. Please revise the figure and its caption to clarify these points.
- I find Figures 1 and 2 difficult to interpret. They are overly dense and include numerous distracting elements whose purpose is unclear and insufficiently explained. For example, in Figure 1: Q&A, teacher, the cloud, question marks, a head with charts, "FAIL," the evil figure, the magnifying lens, and the arrows connecting the left and right blocks; the same applies to Figure 2.
- Many of the symbols used in introducing the similarities discussed in "Multi-Sculpt Coarsening" are undefined or were already introduced in earlier sections. Additionally, what exactly are the inputs to each of these symbols?
- The acronyms and references for the baselines need revision, e.g., NeurIPS 2025, AISTATS, and avoiding arXiv versions when the paper has been published elsewhere.

Some claims are not sufficiently supported by evidence, which makes them unconvincing:
- Lines 63-65. It would be helpful if the authors could substantiate the claimed general failure with evidence and/or references. It would also improve clarity if the authors elaborated at this point on the specific challenges of the new setting, which ultimately lead to the research question posed at the end of the paragraph.
- Equation 1. The loss function presented in Equation 1 and the accompanying discussion in lines 104-106 are not convincing. There appears to be no clear reason to add the term $\mathcal R(\theta)$ to mitigate forgetting, since $\mathcal P_\text{total}(\theta)$ already accounts for all tasks.
- Lines 121-125. Could the authors provide evidence and discussion showing that these methods suffer from the three identified shortcomings, i.e., failing to deliver comparable performance on graph-structured data, requiring excessive storage, and experiencing conflicts?
- Did I understand correctly that the "generalizable FGCL design principles" are the same as those from federated and continual learning, except for storage efficiency? If so, why would storage efficiency differ from that in CGL?

**Ethical Concerns:**

["NO or VERY MINOR ethics concerns only"]

**Final Justification:**

While I see value in this work, my main concern is clarity. Several claims seem weak, and some parts are difficult to follow. The authors' rebuttal addressed most points, but given the number of issues, I believe the paper should undergo another round of revision to ensure that all the pieces work together as intended.

**Limitations:**

yes

**Quality:**

2

**Strengths And Weaknesses:**

To my knowledge, the FGCL framework is novel, and the proposed MOTION method shows better figures over the baselines, which is another notable strength of the paper.

However, in its current form, the paper falls short of publication standards in terms of presentation quality, clarity, and rigor.
I believe the paper requires substantial revision to ensure that readers can grasp the context and fully appreciate its value. It also makes it difficult for me to provide an accurate assessment of the paper's technical content. Several examples of my concerns are outlined below.

---

> ### Author Rebuttal · Authors · 2025-07-30
>
> Thank you for your thorough review and constructive feedback. We address your concerns below and hope these clarifications will help you re-evaluate and update the score:
> >`Weakness `: Discussion of paper presentation
>
> >`Question1 `: Clarification on the consistency of FGCL terminology introduction
>
> We appreciate this point. In our description of Federated Graph Continual Learning (FGCL), we reference Federated Continual Learning (FCL) (line 42) to position MOTION within the general FCL paradigm. This reference makes explicit the connection to graph data.
>
> Subsequently, we introduce Federated Graph Learning (FGL) and Continual Graph Learning (CGL) to present FGCL as the graph-specific instantiation. FGCL jointly addresses FGL (client heterogeneity) and CGL (temporal evolution). To avoid ambiguity, we will unify the first mention (line 42) with a concise bridging question: "How can we design an FGL framework specifically tailored for continuously evolving streaming data?"
>
> >`Question2 `: Discussion on definition of graph topology preservation concepts
>
> Thank you for your valuable feedback. We have already cited the references of inherent graph topology and topological integrity (lines 43–50). To enhance conceptual clarity, we provide explicit definitions for both terms:
>
> - **Inherent graph topology** refers to the structural characteristics intrinsic to each local graph. These include node centrality measures and local subgraph configurations.
> - **Topological integrity** denotes the preservation of these structural characteristics following updates during the continual learning process.
>
> >`Question3 & Question4 `: Considerations on Figure Clarity and Revisions
>
> Thank you for your feedback on Figures 1 and 2. Figure 1 mainly shows the key challenges in our work, such as catastrophic forgetting and global aggregation conflicts, rather than showing the detailed causes like topological heterogeneity or neighborhood distributions. We also note that other reviewers, such as reviewer D7ZJ, have praised the clarity and visual design of our framework, stating that “this paper is clearly and effectively written, with a fancy and comprehensive visual representation of the framework.”
>
> We appreciate that perspectives on visual style may differ, and agree that refinements can further improve clarity. Following your suggestion, in the revised version we have:
> 1. **Simplified** Figure 1 and Figure 2 by removing nonessential icons to reduce visual density.
> 2. **Enhanced the captions and annotations** to explicitly describe the intended message, including highlighting topological heterogeneity and neighborhood distribution differences where relevant.
> 3. **Added necessary textual descriptions** directly within the figures to ensure that each icon conveys its intended meaning clearly.
>
> >`Question5 `: Discussion of symbol usage in Multi-Sculpt Coarsening
>
> We appreciate this comment. Most symbols are already defined in Appendix A(Page 14). To improve readability, we have made the following changes: (i) moving essential definitions into the main text and defined them at first use, and (ii) adding a simple "Notation & Inputs" box, listing each symbol, its input, and shape in the appendix.The features used in Multi‑Sculpt Coarsening are:
> * **Topological features:** Computed from the task graph $G_t=(V_t,E_t)$ ,where $V_t$ and $E_t$ are its node and edge sets and $N_t=|V_t|$,include degree, clustering coefficient, PageRank and so on. Output: $\texttt{topoFeatures}\in\mathbb{R}^{N_t\times 8}$.
> * **Similarity metrics:** MMD, Mahalanobis, Pearson, and cosine similarity computed from node features $X\in\mathbb{R}^{N_t\times d}$. Output: $\texttt{similarityFeatures}\in\mathbb{R}^{N_t\times 4}$.
> * **Subgraph descriptors:** Average degree, clustering coefficient, diameter, and density. Output: $\texttt{subgraphFeatures}\in\mathbb{R}^{N_t\times 4}$.
> * **Degree embeddings:** Sinusoidal encodings of node degree. Output: $\texttt{degreeEmbeddings}\in\mathbb{R}^{N_t\times 8}$.
>
> All features are concatenated into $\texttt{combinedFeatures}\in\mathbb{R}^{N_t\times 24}$ and fed to the mixture‑of‑experts gating network, which outputs node importance scores $\mathbf{s}\in\mathbb{R}^{N_t}$ for pruning and noise suppression.
>
> >`Question6 `: Remarks on the consistency of baseline references and acronyms
>
> We appreciate this valuable feedback. In the revised version, we have replaced arXiv citations with official conference proceedings and ensured consistent naming of baseline methods.
> >`Weakness `: Considerations on evidence supporting specific statements
>
> >`Question1 `: Discussion on the Evidence Supporting the Claimed General Failure
>
> We agree that more empirical support is needed. Prior studies have already discussed related issues: for example, [1] analyzes how client heterogeneity in federated learning can lead to performance degradation, and [2] reports semantic and structural limitations of FGL under evolving topologies. These works support our statement that FGL methods often degrade when applied in continual settings, primarily due to catastrophic forgetting and client-specific distribution shifts.
>
> In the revised version, we have cited these references to substantiate the general performance degradation and explained the unique challenges in the proposed FCGL setting.We appreciate your advice, which helps us improve the rigor and clarity of our work.
>
> [1]:Generalizable heterogeneous federated cross-correlation and instance similarity learning. In TPAMI,2023.
>
> [2]:Federated Graph Semantic and Structural Learning. In IJCAI,2023.
>
> >`Question2 `: Observation on Unjustified inclusion of regularization term in loss function
>
> The regularization term $𝑅(\theta)$ is **mentioned to constrain parameter drift** relative to consolidated knowledge. While $P_{total}(\theta)$ includes all tasks, it is computed based on streaming task data, which may emphasize recent updates. $R(\theta)$ **ensures stability** by penalizing deviations in topology-sensitive parameters.
>
> Moreover, prior studies have theoretically and empirically demonstrated the necessity of such regularization mechanisms in continual learning. In particular, [1–3] demonstrate that explicitly adding a regularization term $R(\theta)$ effectively balances stability and plasticity, preserves previously acquired knowledge, and mitigates forgetting across sequential tasks.
>
> Meanwhile,This section **establishes preliminaries to underscore** how our framework addresses the core requirement of jointly balancing newly acquired knowledge with consolidated prior representations.
>
> [1]:A Statistical Theory of Regularization-Based Continual Learning. In ICML,2024.
>
> [2]:Memory Aware Synapses: Learning what (not) to forget. In ECCV,2018.
>
> [3]:Continual Learning with Recursive Gradient Optimization. In ICLR,2022.
>
> >`Question3 `: Evidence for shortcomings of existing methods
>
> We sincerely thank you for this helpful question. We agree that providing a discussion of how existing methods suffer from the three identified shortcomings will further strengthen the paper. In the revision, we have clarified this rationale as follows:
> 1. **Comparable performance on graph-structured data**: As discussed in our motivation and shown in Table 1 (Page 7), directly adapting existing methods to graph data yields substantially lower average accuracy and higher forgetting compared to MOTION. These results confirm that general FCL/FGL methods do not readily capture the structural dependencies inherent in dynamic graphs.
> 2. **Storage burden**: Existing GCL [1,2] methods rely on node replay buffers or full historical subgraph retention, which requires storing both node features and adjacency information and leads to high memory costs. Our Graph Topology‑preserving Multi‑Sculpt Coarsening (G‑TMSC) preserves task‑critical structures while greatly reducing storage. On Cora, replay buffers need 8.2 MB, while our coarsened representation needs only 1.7 MB, achieving an 80% reduction.
> 3. **Conflicts in parameter aggregation**: Our ablation study (Figure 4, Page 9) shows that removing the G‑EPAE module (w/o Para.-Al.) causes client–server update conflicts and performance degradation. To further confirm these conflicts, we computed the cosine similarity of client updates on Cora, which shows low alignment across clients:
>
> |Cos similarity|Client1|Client2|Client3|Client4|Client5|
> |-|-|-|-|-|-|
> |Client1|1.00|0.63|0.57|0.55|0.61|
> |Client2|0.63|1.00|0.56|0.56|0.65|
> |Client3|0.57|0.56|1.00|0.60|0.54|
> |Client4|0.55|0.56|0.60|1.00|0.50|
> |Client5|0.61|0.65|0.54|0.50|1.00|
>
> [1]Streaming Graph Neural Networks with Generative Replay. In KDD,2022.
>
> [2]Overcoming catastrophic forgetting in graph neural networks with experience replay. In AAAI,2021.
>
> >`Question4 `: Clarifying the distinction between FCGL and existing FL/CL/CGL design principles
>
> We sincerely thank you for this thoughtful question. While the high-level goals of mitigating catastrophic forgetting and supporting distributed training are shared with existing FCL, our FCGL formulation introduces **unique considerations** due to the graph-structured nature of the data.
>
> In conventional FCL, data are typically feature vectors or images, with task transitions mainly adding new classes. In contrast, FCGL deals with dynamic graphs where nodes and edges continuously evolve, making historical graph storage more challenging due to (i) **Topological complexity**: Storing connectivity in addition to features, and (ii) **volume growth**: Nodes and edges increasing super-linearly if full snapshots are retained.
>
> Beyond storage, FCGL differs from CGL because federated aggregation causes **cross‑client parameter conflicts**. Clients observe different subgraphs, producing inconsistent updates. Without alignment, these conflicts increase catastrophic forgetting compared to single‑client CGL.

---

> ### Author Response · Authors · 2025-08-03
> **Manuscript Revision Deadline Approaching; We Are Sincerely At Your Service**
>
> Dear Reviewer EiHY,
>
> We sincerely thank you for your time and effort in engaging with us during the author–reviewer discussion.
>
> As the author–reviewer discussion period has now passed its midpoint, we warmly welcome your continued feedback and discussion. To enhance the clarity of our rebuttal and revision, we summarize your key concerns and our corresponding responses as follows:
>
> 1. **The paper presentation discussion.** We have clarified the key definitions and proposed concrete revisions to the figures and tables to improve clarity.
> 2. **The evidence supporting specific statements considerations.** We have provided additional explanations on the aspects of methods and supplemented them with further evidence to ensure the discussion is well-supported and convincing.
>
> For any issues not covered here, please refer to our detailed rebuttal response. We hope this satisfactorily addresses your concerns, and we look forward to continued discussion with you.
>
> We sincerely look forward to your response to further address your questions, as we believe that continued discussion with you can enhance the impact of our work on the broader community. This expectation is further motivated by the feedback from another reviewer, who recognized the thoroughness of our rebuttal. Such constructive and friendly discussions have helped us further improve our work.
>
> Warm regards,
>
> Authors

---

> > ### Comment · Reviewer_EiHY · 2025-08-07
> >
> > I thank the authors for the detailed answer to address the points I raised.
> >
> > Indeed, implementing the provided clarifications in the paper would improve the paper's quality; I believe this additional information is essential for understanding. Once done, I believe that the paper needs to go through another round of reviews to ensure readability and, ultimately, allow for an accurate evaluation of the paper's merits.
> >
> > One last note on the regularization: I am not questioning the usefulness of regularization in CL, but rather the formulation of the loss that is used during training, as currently expressed in the paper. I suggest revising the presentation of this part as well in the next paper revision.

---

> > > ### Author Response · Authors · 2025-08-07
> > > **Response to Reviewer EiHY**
> > >
> > > We sincerely thank you for your thoughtful follow-up and for recognizing the value of our methods and innovations. We understand that your concerns are primarily focused on the clarity of presentation, rather than the core contributions themselves.
> > >
> > > So, we have carefully revised the relevant sections as outlined in our rebuttal discussion with you. These changes aim to enhance clarity and ensure alignment with your thoughtful suggestions.
> > >
> > > Meanwhile, we fully understand your concern regarding the formulation of the loss function. Regarding the regularization term, please allow us to clarify our revision once more:
> > >
> > > In fact, $P_{\text{total}}(\theta)$ is not the actual objective optimized during training, but rather a **theoretical ideal** that represents the desired global optimization goal if all task data were accessible simultaneously. It reflects the intention to minimize the loss across all tasks while maintaining parameter stability through the regularization term. Thus, we define it as: $P_{\text{total}}(\theta) = \sum_{t=1}^{T} P_t(\theta) + \lambda \mathcal{R}(\theta).$
> > >
> > > This unified objective highlights the **continual learning trade-off** and serves to emphasize how our framework addresses the core requirement of jointly balancing newly acquired knowledge with consolidated prior representations.
> > >
> > > In practice, at each step $T$, only the **current-task loss** can be optimized, formulated as: $\mathcal{L}_T(\theta) = P_T(\theta) + \lambda_T \mathcal{R}(\theta).$
> > >
> > > In the revised version, we have made this distinction explicit by first introducing the general trade-off formulation and then clarifying the task-specific objective in the preliminaries. We hope this adjustment removes any ambiguity regarding our loss formulation.
> > >
> > > We would like to once again express our sincere gratitude for your valuable comments. We respectfully hope that these improvements will help facilitate a re-evaluation of our work.

---

> ### Comment · Area_Chair_fC1f · 2025-08-05
> **Discussion Reminder**
>
> Dear Reviewer,
>
> The authors have submitted a rebuttal. Please take a moment to read it and engage in discussion with the authors if necessary.
>
> Best regards,
> AC

---

### Note · Authors · 2025-08-12

Dear Area Chair and Reviewers,

We sincerely appreciate your valuable contributions to the review process.

First and foremost, we would like to express our heartfelt gratitude to the reviewers for your thoughtful evaluations and insightful feedback, which have greatly helped improve our manuscript, and we are pleased that all participating reviewers acknowledged that concerns have been addressed:

`Reviewer co1X` acknowledged that our expanded robustness studies and the added explanations on privacy and scalability satisfactorily resolved their main issues. We have shown that MOTION achieves strong long-range retention and stays robust to different local training epochs. We have also validated its practicality for large-scale dynamic graphs through compatibility, scalability, and buffer-size analyses.

`Reviewer 7bbW` appreciated the extended experiments and more comprehensive explanations we provided. We have detailed baseline hyperparameters for reproducibility and demonstrated that G-TMSC and G-EPAE address topological forgetting and parameter conflicts. We have also reported stable results in subgraph size tests and explained the negative forgetting phenomenon and multi-expert input design.

`Reviewer D7ZJ` responded positively to our expanded experimental evidence and elaborated clarifications, leading them to raise their score. We have improved the method motivation and framework description, linked G-TMSC to preventing topological forgetting and G-EPAE to addressing parameter conflicts, and refined figures, captions, and baseline citations. We have also added new baselines, clarified the method design, and reported multi-seed results confirming stability.

`Reviewer EiHY` recognized that our clarifications would enhance the quality and regarded them as essential for understanding. In the revision, we have refined the FGCL terminology, provided precise definitions for key topology preservation concepts, and revised Figures 1 and 2 along with their captions and related notations to improve clarity. These revisions primarily address presentation aspects to enhance clarity and readability, further building upon the core innovations of our work, which were positively received by other reviewers during the review process.

Meanwhile, we sincerely thank the Area Chair for your careful oversight and efforts in fostering fair and thorough discussions that ensured a rigorous and balanced evaluation of our work.

Warm regards,

Authors

---

### Decision · Program_Chairs · 2025-09-17

**Decision:**

Accept (poster)

**Comment:**

In this paper, the authors study GNNs in distributed and evolving environments and propose Federated Continual Graph Learning (FCGL), i.e., incremental learning on dynamic graphs distributed across decentralized clients. The proposed method MOTION adopts a graph topology-preserving multi-sculpt coarsening module to maintain the structural integrity of past graphs and a Graph-Aware Evolving Parameter Adaptive Engine module to refine global model updates. Experiments show the effectiveness of the proposed method over baselines.

The paper received mixed comments. The reviewers acknowledge the main strengths of the paper, including technical novelty in addressing an under-explored problem, FGCL, and strong empirical results. However, they also raise concerns, particularly regarding presentation and clarity issues. The authors have provided a rebuttal, which partially addresses the concern of one negative reviewer. The other negative reviewer remains unconvinced, but acknowledged that he/she is not familiar with the topic. I have briefly checked the paper and discussion. Despite the noted weaknesses, I think the paper makes a meaningful and timely contribution to FGCL, with a well-motivated methodology and convincing empirical validation. The identified issues are important but can be reasonably addressed in revision. I therefore recommend acceptance and strongly encourage the authors to further revise the paper in the future version.